ecology

population dynamic, population density, principal components analysis, quartile method

**Authors for correspondence:**
Guohua Chen
e-mail: chenghkm@126.com
Xiaoming Zhang
e-mail: zxmalex@126.com

# Regulation of dynamics and densities of whitefly *Bemisia tabaci* by agricultural landscapes in south China

Shaowu Yang, Wenjun Dou, Mingjiang Li, Xingxing Li, Zhengxiong Jiang, Guohua Chen and Xiaoming Zhang

State Key Laboratory for Conservation and Utilization of Bio-Resources in Yunnan, College of Plant Protection, Yunnan Agricultural University, Kunming 650201, People's Republic of China

SY, 0000-0001-9046-9088; GC, 0000-0003-3696-6416; XZ, 0000-0002-2753-2408

Agricultural landscape patterns can affect the population dynamics of pest insects. We selected four landscapes (flower field, mountain, river and urban) based on principal components analysis in Yunnan Province, south China. Through systematic investigation carried out in tomato fields, we intended to clarify the population dynamics and densities of *Bemisia tabaci* under different landscape types. During the main activity period of *B. tabaci*, the population densities of *B. tabaci* nymphs in tomato fields in the river and the urban landscape types were the highest compared to the other landscape types; the population densities of female adults in tomato fields in the river landscape type were also the highest. While the population densities of *B. tabaci* nymphs and female adults in the flower landscape type, no more than five individuals (ind.) 100 cm$^{-2}$ leaf in both years, were the lowest. The density of *B. tabaci* nymphs in the middle position of tomato plants was higher than those in the other positions, while the density of adults in the upper position of tomato plants was higher, regardless of landscape types. Our findings showed that the population growth of *B. tabaci* can be easily controlled by the flower landscape type.

## 1. Introduction

With the intensification and scale of agricultural production, agricultural landscapes in most parts of the world have changed from a complex pattern with a large proportion of natural habitats to a simple landscape with a large proportion of arable land, which leads to the rapid decline of biodiversity in agroecosystems [1–3]. The change in agricultural landscape

pattern can affect the population densities of pests [4–6]. On the one hand, some studies have shown that richer diversity of the agricultural landscape is helpful to reduce the damage from pests [7,8]. The presence of diverse non-crop habitats in structurally complex landscapes may greatly influence the activity and abundance of natural enemies, including parasitoids and predators [9,10]. These diverse non-crop habitats can provide food, shelter, alternative hosts, and favourable microclimates for natural enemies in the field, which supports pest control [1,11]. On the other hand, landscape diversity is conducive to the survival of pests [12]. Diversity of plants, including non-crops and weeds, not only provides a good living place for different ages or states of pests but also provides sufficient food for polyphagous pests [13,14].

The whitefly, *Bemisia tabaci* (Gennadius) (Hemiptera: Aleyrodidae), is one of the most economically and agriculturally important insect pests worldwide [15,16]. It is distributed in more than 90 countries and regions around the world and is considered to be polyphagous in nature with more than 1000 plant species listed as hosts including vegetables, flowers, cotton and tobacco [17–19]. The nymphs and adults of *B. tabaci* can cause economic damage directly by sucking plant fluids and indirectly by transmitting plant viruses including *Begomovirus*, *Carlavirus*, *Ipomovirus* and *Torradovirus* [20,21].

In our previous study, we found that the damage degrees of *B. tabaci* in tomato fields were different in different landscape types around Kunming, Yunnan Province (unpublished data). Therefore, we hypothesized that the different agricultural landscape types may affect the dynamics and densities of whitefly in tomato fields. Yunnan Province is one of the regions with the richest biodiversity in the world [22,23]. However, due to the development of the agricultural economy and the intensification of human activities, the agricultural landscape of Yunnan Province has changed significantly, which has broken the original ecological balance and led to the outbreak of whitefly [24,25]. In this study, we selected four typical landscape types, with flower fields, mountain, river and urban areas as the main elements in central Yunnan, where agricultural activities are most frequent in the plateau agricultural planting areas of Yunnan Province. Through systematic investigation, we intended to clarify the population dynamics and spatial distribution of *B. tabaci* under different landscape types. The aim of this study, therefore, was to find out which landscape type is most conducive to decreasing the population level of *B. tabaci* to provide the scientific basis for the ecological management of *B. tabaci* and establish a green eco-agricultural landscape with sustainable development for many years.

# 2. Materials and methods

## 2.1. Study area

The study was conducted at 12 tomato field plots (20 m × 40 m) among 12 agriculture landscapes with radius of 0.5 km, which were located in the surroundings of Kunming, south China (24°42′45″ N 25°22′43″ N, 102°22′18″ E-103°10′90″ E) (figure 1). The land cover types were selected using Google Earth Profession and field inspections (ground-truthing) once a month during the tomato growing seasons in 2018 and 2019 [26–28]. The cover types in each landscape were divided into 10 types according to vegetation type, human factor interference and land type characteristics: (i) Flower fields, (ii) River, (iii) Mountain (cypress forest with altitude difference more than 150 m), (iv) Urbans, (v) Vegetable fields, (vi) Fruit trees, (vii) Trees (windbreaks, border trees or ornamental trees), (viii) Bushes, (ix) Grasslands and (x) Wastelands. The altitude difference among the landscape types was within 20 m, except for mountains. A principal components analysis (PCA) was performed to reduce the dimensions of the landscape data. These 10 land cover types were divided for the PCA analysis; the land cover type with the largest area in one landscape and the absolute value of the first principal component greater than 0.9 was selected as the landscape type. Principal component axes were extracted using correlations among variables, and the resulting factors were not rotated. We divided the 12 landscapes into four types according to the interpretation of principal components (electronic supplementary material, data S1):

  (i) Flower landscape type: three of them were divided into flower landscape type, their main landscape cover types were flower fields, which were evenly distributed in the agricultural landscape types, and the main species of flower were *Rosa chinensis*, *Dianthus caryophyllus*, *Myosotis sylvatica* and *Eustoma grandiflorum*.
  (ii) River landscape type: three of them were divided into river landscape type, their main landscape cover types were river. The Panlong river across these three landscapes; the purpose of setting up the river landscape type was to pay attention to the high humidity environment caused by the river.

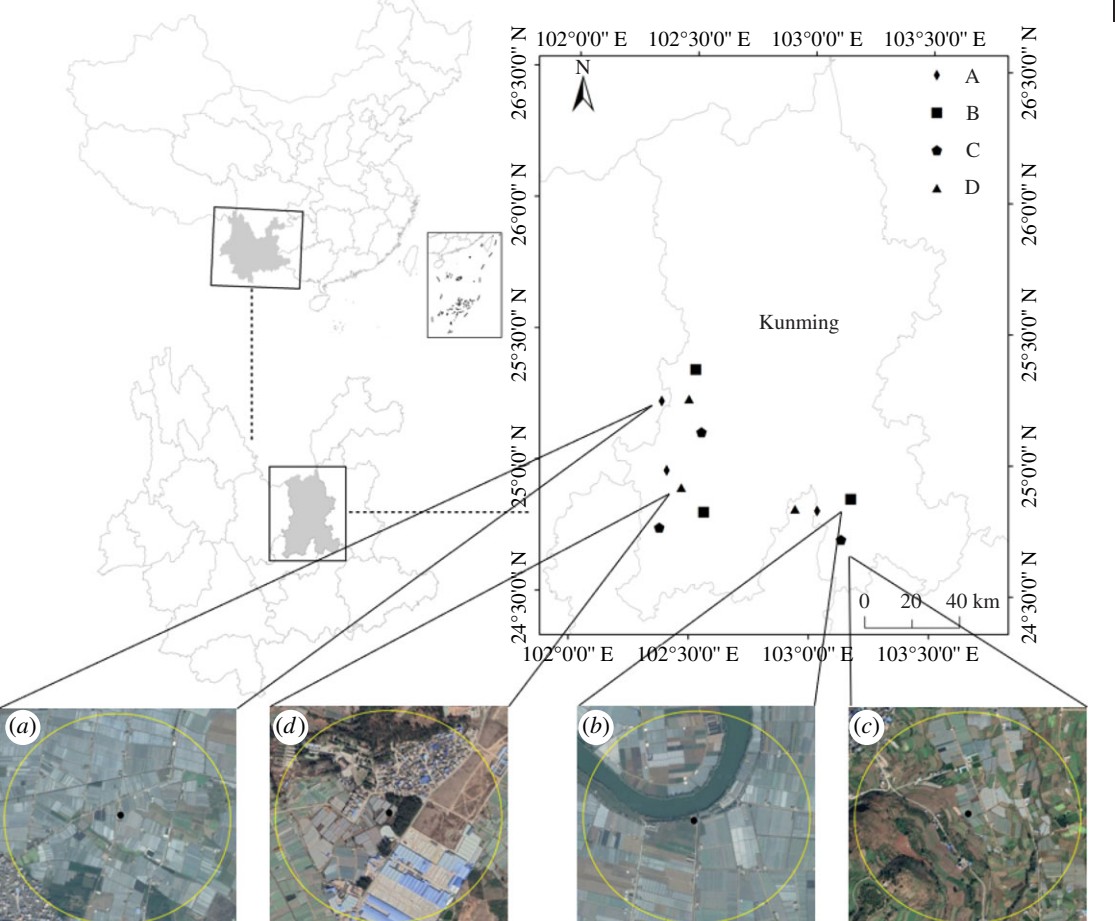

**Figure 1.** Site locations of study area in Kunming, south China. Note: (*a*) landscape dominated by flower fields, (*b*) landscape dominated by river, (*c*) landscape dominated by mountain and (*d*) landscape dominated by urban areas.

(iii) Mountain landscape type: three of them were divided into mountain landscape type; their main landscape cover types were cypress forest with altitude difference more than 150 m.
(iv) Urban landscape type: three of them were divided into urban landscape types. These landscapes were close to the town and their main landscape cover types were buildings.

Each plot was at least 5 km apart, and there were no two plots of the same landscape type adjacent to each other. The study was carried out in tomato planting fields in 2018 and 2019 under different landscape types. The cultivar planted was tomato cv. 'Zhongyan TV1' (*Lycopersicon esculentum* Mill.). No plot was treated against insect or disease pests in our experiments. Plots were kept weed-free by manual weeding as necessary.

## 2.2. Sampling

In each 800 m$^2$ plot (field plots: 20 m × 40 m; plant spacing: 40 cm; row spacing: 80 cm) of the tomato fields in each landscape, the first sampling started 10 days after tomato transplanting between Jun. and Nov. in 2018 and 2019. Leaves with *B. tabaci* nymphs and adults were collected every 10 days using the fixed five points sampling method until the end of the tomato growing season. In each sampling point, five tomato plants were sampled (but avoiding the plants closest to any edge to minimize edge effects). On each tomato plant, five leaves of similar age at the upper, middle and lower positions were examined, giving a total of 375 leaves observed per plot. Adults were counted *in situ*; afterwards, the leaves were cut, individually placed in plastic bags and marked, then brought to the laboratory to count nymphs under a dissecting microscope (20 × magnification, MOTIC, SMZ-168). The area of each leaf was measured by a leaf area measuring instrument (Yaxin-1242), from which

standardized density data (no. of individuals per 100 cm$^2$ leaf surface) were obtained. The tomato growth period was recorded at each survey. This was divided into seedling, anthesis, fruit expansion and harvest periods [29,30].

## 2.3. Describing seasonal activity

The seasonal activity curve was standardized following the quartile method of Fazekas *et al.* and Zhang *et al.* [24,31]. This method divides the seasonal activity into three periods—early, main and late—and formally identifies the start and end for each of these, as well as the date of the seasonal activity peak. First, the numbers observed were summed and the three cardinal points were the dates when 25, 50 and 75% of the total densities were reached. These also divided the curve into four segments. The start of the main activity period was from the date when the cumulative densities reached 25% of the total (the start of the second quartile on the vertical axis), and the end was the date when 75% (the end of the third quartile on the vertical axis) was reached. The formally defined early activity period extended from the start of the census to the beginning of the main activity period, and the likewise formalized late activity period lasted from the end of the main activity period until the end of the observations, when activity stopped.

## 2.4. Tomato yield

Four spikes of tomato were reserved in the tomato plants in the fields. To obtain data on tomato yield, the five-point sampling method was used. On each plot, five tomato plants were randomly selected during each picking. The mature tomatoes were harvested and taken back to the laboratory to be weighed. The yield per unit area was calculated according to the average yield per plant and the plant and row spacing of tomato plants in the tomato fields [32,33].

## 2.5. Data analysis

Census data were initially subjected to a one-way ANOVA (repeated measures) after tests of normality (Shapiro–Wilk) and homoscedasticity (Bartlett), with agriculture landscape types as the main effect. To reduce the impact of occurrence time on the population densities of *B. tabaci*, the activity period of *B. tabaci* was divided into early, main and late activity period by quartile method. Differences in *B. tabaci* densities were compared among agriculture landscape types in the same activity period, as well as densities in different positions of the tomato plants, constrained by landscape type (identified as above) by the least significant difference. The significance threshold was $p = 0.05$ in all tests. Data analyses were performed using SPSS 20.0. The figures of population dynamics and cumulative seasonal activity curves were created using Origin 2018.

# 3. Results

## 3.1. Seasonal activity of *Bemisia tabaci* nymphs and adults under different landscape types

In 2018, the main activity period of *B. tabaci* nymphs in the flower landscape type, as long as 52 days, was longer as compared to other landscape types. The peak activity date in the tomato fields in the river landscape type, 28 Aug., was the earliest relative to the other landscape types, and that in the flower and the urban landscape types, 10 Sept. and 19 Sept., respectively, were later relative to the other landscape types. In 2019, the peak activity date of *B. tabaci* nymphs in tomato fields in the flower landscape type was 14 Sept., and its main activity period, as long as 65 days, was the longest as compared to other landscape types. The main activity periods of *B. tabaci* nymphs in the urban and the river landscape types ranged from 30 Aug. to 20 Sept., as long as 22 days, and the peak activity date was in early September. The main activity period of *B. tabaci* nymphs in the mountain landscape type, as long as 22 days, was the same relative to that in the river and the mountain landscape types, but the peak activity date, 26 Aug., was the earliest relative to that in the other landscape types (table 1 and figure 2).

In 2018, the main activity period of female adult *B. tabaci* in the flower landscape type, as long as 43 days, was the longest as compared to other landscape types. The peak activity date in the tomato fields in the flower landscape type, 30 Aug., was the earliest relative to the other landscape types, and that in the urban landscape type, 26 Sept., was the latest relative to the other landscape types. In 2019, the main

**Table 1.** Main activity periods and peak activity dates of *Bemisia tabaci* nymphs and adults under the different landscape types in Kunming, south China. Note: A: landscape dominated by flower fields; B: landscape dominated by river; C: landscape dominated by mountain; D: landscape dominated by urban areas.

| planting years | | landscape types | main activity period (duration in days) | peak activity date |
|---|---|---|---|---|
| 2018 | nymphs | A | 08 Aug.–28 Sept. (52) | 10 Sept. |
| | | B | 21 Aug.–10 Sept. (21) | 28 Aug. |
| | | C | 29 Aug.–17 Sept. (20) | 03 Sept. |
| | | D | 15 Sept.–27 Sept. (13) | 19 Sept. |
| | female adults | A | 08 Aug.–19 Sept. (43) | 30 Aug. |
| | | B | 30 Aug.–10 Sept. (12) | 06 Sept. |
| | | C | 07 Sept.–17 Sept. (11) | 12 Sept. |
| | | D | 15 Sept.–08 Oct. (24) | 26 Sept. |
| | male adults | A | 28 Jul.–19 Sept. (54) | 28 Aug. |
| | | B | 30 Aug.–19 Sept. (21) | 09 Sept. |
| | | C | 29 Aug.–29 Sept. (32) | 12 Sept. |
| | | D | 02 Sept.–27 Sept. (26) | 26 Sept. |
| 2019 | nymphs | A | 08 Aug.–10 Oct. (65) | 14 Sept. |
| | | B | 30 Aug.–20 Sept. (22) | 04 Sept. |
| | | C | 20 Aug.–10 Sept. (22) | 26 Aug. |
| | | D | 30 Aug.–20 Sept. (22) | 10 Sept. |
| | female adults | A | 20 Aug.–10 Oct. (52) | 18 Sept. |
| | | B | 10 Sept.–20 Sept. (11) | 16 Sept. |
| | | C | 20 Aug.–10 Sept. (22) | 02 Sept. |
| | | D | 10 Sept.–20 Sept. (11) | 16 Sept. |
| | male adults | A | 28 Jul.–30 Sept. (65) | 14 Sept. |
| | | B | 10 Sept.–20 Sept. (11) | 16 Sept. |
| | | C | 08 Aug.–20 Sept. (44) | 12 Sept. |
| | | D | 30 Aug.–20 Sept. (22) | 16 Sept. |

activity period of female adult *B. tabaci* in the flower landscape type, as long as 52 days, was the longest as compared to other landscape types, and the peak activity date, 18 Sept., was the latest relative to the other landscape types. The main activity period of female adult *B. tabaci* in the river and the urban landscape types ranged from 10 Sept. to 20 Sept., as long as 11 days, and their peak activity date was in mid-Sept. (table 1 and figure 2).

In 2018, the main activity period of male adult *B. tabaci* in the flower landscape type, as long as 54 days, was the longest relative to the other landscape types. Its peak activity date, 28 Aug., was earliest relative to the other landscape types. The main activity period of male adult *B. tabaci* in the river landscape type, as long as 21 days, was the shortest relative to that in the other landscape types. In 2019, the main activity period of male adult *B. tabaci* in the flower landscape type, as long as 65 days, was the longest relative to the other landscape types, and it peaked on 14 Sept., which in the river landscape type, as long as 11 days, was shortest relative to the other landscape types, and it peaked on 16 Sept. (table 1 and figure 2).

## 3.2. Population dynamics of *Bemisia tabaci* nymphs and adults under different landscape types

In the flower landscape type, the *B. tabaci* nymphs and adults were maintained at low levels, and the population densities were no more than 10 ind. 100 cm$^{-2}$ leaf during the whole sampling period in the tomato fields in both years (figure 3a).

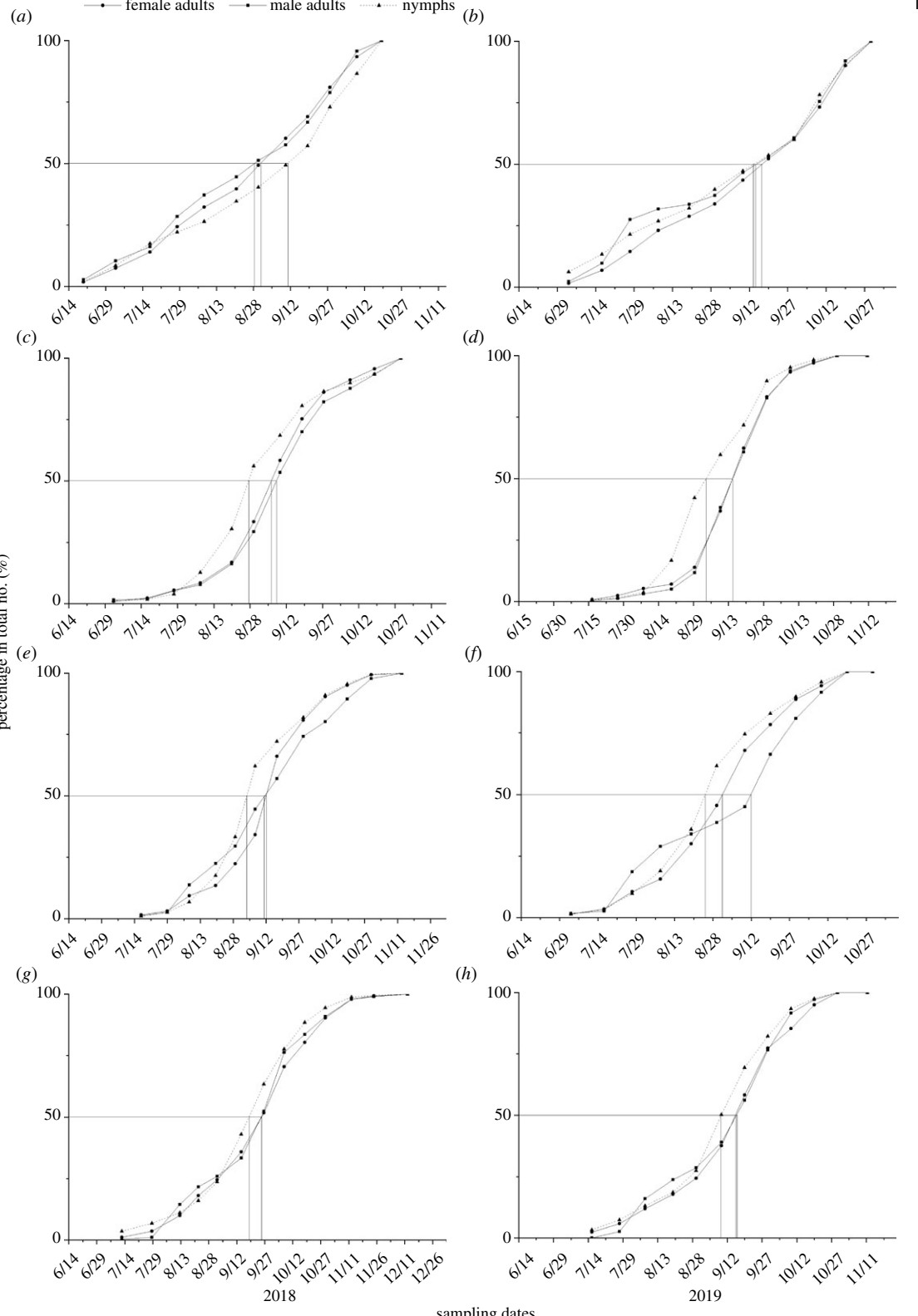

**Figure 2.** Cumulative seasonal activity curves of *Bemisia tabaci* nymphs and adults under the different landscape types in Kunming, south China. Note: (*a,b*) landscape dominated by flower fields, (*c,d*) landscape dominated by river, (*e,f*) landscape dominated by mountain and (*g,h*) landscape dominated by urban areas.

In the river landscape type, the densities of female adults of *B. tabaci* in tomato fields increased slowly in the early activity period and began to increase rapidly around early to mid-Aug. in both 2018 and 2019. The densities of female adult *B. tabaci* peaked at 79.42 and 87.00 ind. 100 cm$^{-2}$ leaf in mid-Sept.,

respectively, and then decreased sharply until the end of the sampling. The densities of *B. tabaci* nymphs in tomato fields increased slowly in the early activity period and began to increase rapidly around late Jul. to early Aug. in both years. They peaked at 85.45 and 90.40 ind. 100 cm$^{-2}$ leaf in the end of Aug., respectively. After that, the densities of nymphs in 2018 decreased to mid-Oct. and then there was a small rise again; the densities of nymphs in 2019 decreased to the end of Sept. and reached 63.56 ind. 100 cm$^{-2}$ leaf in early Oct., then decreased sharply until the end of the sampling. The density of male adult *B. tabaci* maintained a low level in the early activity period and gradually increased to mid and late Aug. to the first 10 days of Sept., and then the densities began to decline slowly to the end of sampling (figure 3*b*).

In the mountain landscape type, the densities of nymphs and female adults *B. tabaci* in tomato fields increased slowly at the beginning of the sampling. The densities of nymphs peaked at 50.74 and 39.11 ind. 100 cm$^{-2}$ leaf at the end of Aug. to early Sept., respectively, and the female adults peaked at 55.94 and 36.00 ind. 100 cm$^{-2}$ leaf in early to mid-Sept., respectively. After that, the densities of nymphs and female adults decreased to the end of the sampling. The density of male adult *B. tabaci* maintained a low level during the whole sampling period in both 2018 and 2019 (figure 3*c*).

In the urban landscape type, the population densities of nymphs and female adults *B. tabaci* increased gradually at mid-Jul. The *B. tabaci* nymphs peaked at 68.86 and 80.95 ind. 100 cm$^{-2}$ leaf in mid and late Sept., and female adults peaked at 56.30 and 62.81 ind. 100 cm$^{-2}$ in mid and late Sept., respectively. After that, the population densities of nymphs and female adults *B. tabaci* decreased sharply. The nymphs decreased to the end of the sampling, and the female adults decreased to the end of the sampling after a small increase in mid to late Oct. The average densities of male adult *B. tabaci* were less than 10 ind. 100 cm$^{-2}$ during the whole sampling period in both years (figure 3*d*).

The densities of *B. tabaci* nymphs in the river and the urban landscape types were higher in 2018, and their densities reached 62.57 and 66.88 ind. 100 cm$^{-2}$ leaf in the main activity periods, respectively, the highest as compared to other landscape types. At the early and main activity periods, the densities of *B. tabaci* nymphs in the flower landscape type, at 3.09 and 4.75 ind. 100 cm$^{-2}$ leaf, respectively, were significantly lower as compared to other landscape types in 2018 (early activity period: $F_{3,50} = 6.53$, $p = 0.001$; main activity period: $F_{3,41} = 53.07$, $p < 0.001$; late activity period: $F_{3,53} = 2.71$, $p = 0.055$). At the early activity period of *B. tabaci* nymphs in 2019, the density of *B. tabaci* nymphs (16.65 ind. 100 cm$^{-2}$ leaf) was the highest in the urban landscape type and the lowest in the flower landscape type (3.17 ind. 100 cm$^{-2}$ leaf) ($F_{3,44} = 3.82$, $p = 0.017$). The densities of *B. tabaci* nymphs in the river and the urban landscape types, at 65.19 and 60.48 ind. 100 cm$^{-2}$ leaf, respectively, were the highest relative to the other landscape types, and the density in the flower landscape type, only 3.58 ind. 100 cm$^{-2}$ leaf, was the lowest relative to that in the other landscape types ($F_{3,44} = 45.28$, $p < 0.001$). In the late activity period of *B. tabaci*, the densities of *B. tabaci* nymphs in the river and the urban landscape types, at 20.05 and 21.84 ind. 100 cm$^{-2}$ leaf, respectively, were significantly higher as compared to other landscape types in 2019 ($F_{3,59} = 5.14$, $p = 0.003$) (table 2).

The densities of female adult *B. tabaci* in the river landscape type were the highest in 2018 at the main and late activity period, reaching 66.19 and 26.67 ind. 100 cm$^{-2}$ leaf, respectively. At each activity period, the densities of female adult *B. tabaci* in the flower landscape types, at 2.75, 4.06 and 4.66 ind. 100 cm$^{-2}$ leaf, respectively, were significantly lower as compared to other landscape types in 2018 (early activity period: $F_{3,56} = 6.91$, $p = 0.001$; main activity period: $F_{3,35} = 33.77$, $p = 0.0001$; late activity period: $F_{3,53} = 5.65$, $p = 0.002$). At the early activity period in 2019, the density of female adult *B. tabaci* in the urban landscape type, at 14.92 ind. 100 cm$^{-2}$ leaf, was the highest relative to that in the other landscape types, and the density in the flower landscape type, at 2.33 ind. 100 cm$^{-2}$ leaf, was the lowest relative to that in the other landscape types ($F_{3,53} = 12.48$, $p < 0.001$). At the main activity period, the density of female adult *B. tabaci* in the river landscape type, at 82.56 ind. 100 cm$^{-2}$ leaf, was highest relative to that in the other landscape types, and the density in the flower landscape type, at 3.39 ind. 100 cm$^{-2}$ leaf, was the lowest relative to that in the other landscape types ($F_{3,38} = 146.89$, $p < 0.001$). At the late activity period, the densities of female adult *B. tabaci* in the river and the urban landscape types, at 25.64 and 25.54 ind. 100 cm$^{-2}$ leaf, respectively, were higher as compared to the flower and mountain landscape types ($F_{3,56} = 5.57$, $p = 0.002$) (table 2).

At the early and late activity periods, the densities of male adult *B. tabaci* were lower than 10 ind. 100 cm$^{-2}$ leaf no matter what the year. At the main activity period, the densities of male adult *B. tabaci* in the river landscape type (12.62 ind. 100 cm$^{-2}$ leaf in 2018 and 15.95 ind. 100 cm$^{-2}$ leaf in 2019) were higher as compared to other landscape types (in 2018: $F_{3,47} = 19.90$, $p < 0.001$; in 2019: $F_{3,50} = 53.25$, $p < 0.001$) (table 2).

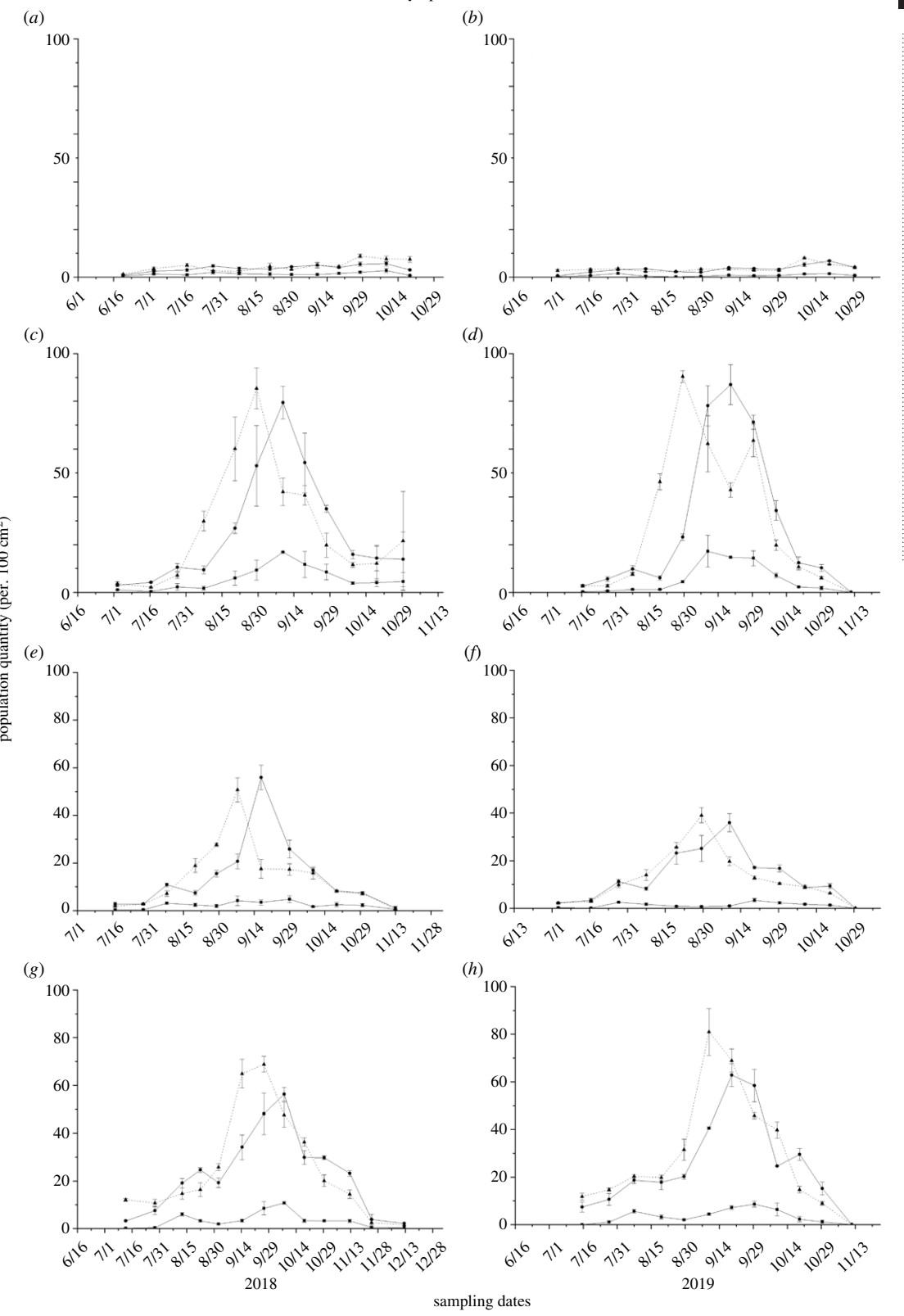

**Figure 3.** Population dynamics of *Bemisia tabaci* nymphs and adults (mean ± s.e.) under the different landscape types in Kunming, South China. Note: (*a,b*) landscape dominated by flower fields, (*c,d*) landscape dominated by river, (*e,f*) landscape dominated by mountain and (*g,h*) landscape dominated by urban areas.

**Table 2.** Population densities (mean ± s.e.) of nymphs and adults of *Bemisia tabaci* in different activity periods under the different landscape types in Kunming, south China. Notes: the different lower case letters indicate a significant difference at the 0.05 level in the different landscape types during the same tomato planting year. A: landscape dominated by flower fields; B: landscape dominated by river; C: landscape dominated by mountain; D: landscape dominated by urban areas.

| planting years | | landscape types | population densities (per. 100 cm$^2$) | | |
| --- | --- | --- | --- | --- | --- |
| | | | early activity period | main activity period | late activity period |
| 2018 | nymphs | A | 3.09 ± 0.47 c | 4.75 ± 0.57 c | 7.54 ± 0.81 abc |
| | | B | 10.75 ± 3.51 ab | 62.57 ± 7.95 a | 21.17 ± 4.79 a |
| | | C | 7.70 ± 2.15 bc | 32.00 ± 5.25 b | 9.78 ± 1.75 abc |
| | | D | 15.94 ± 1.60 a | 66.88 ± 3.20 a | 20.46 ± 4.19 ab |
| | female adults | A | 2.75 ± 0.45 c | 4.06 ± 0.30 c | 4.66 ± 0.58 c |
| | | B | 10.79 ± 2.36 ab | 66.19 ± 10.04 a | 26.67 ± 5.21 a |
| | | C | 7.86 ± 1.36 bc | 38.31 ± 8.33 b | 11.93 ± 2.39 bc |
| | | D | 14.72 ± 2.22 a | 46.18 ± 4.44 b | 17.77 ± 3.36 ab |
| | male adults | A | 0.89 ± 0.19 a | 1.40 ± 0.21 b | 1.84 ± 0.41 b |
| | | B | 2.28 ± 0.78 a | 12.62 ± 2.29 a | 5.27 ± 1.27 a |
| | | C | 1.55 ± 0.41 a | 3.57 ± 0.68 b | 1.78 ± 0.35 b |
| | | D | 2.43 ± 0.75 a | 4.58 ± 1.30 b | 3.56 ± 0.85 ab |
| 2019 | nymphs | A | 3.17 ± 0.24 c | 3.58 ± 0.13 c | 3.04 ± 0.84 b |
| | | B | 14.86 ± 5.56 ab | 65.19 ± 7.76 a | 20.05 ± 6.19 a |
| | | C | 7.21 ± 1.58 bc | 28.18 ± 3.05 b | 7.72 ± 1.19 b |
| | | D | 16.65 ± 1.17 a | 60.48 ± 8.20 a | 21.84 ± 4.81 a |
| | female adults | A | 2.33 ± 0.34 c | 3.39 ± 0.30 d | 3.07 ± 0.80 b |
| | | B | 9.44 ± 1.98 b | 82.56 ± 5.66 a | 25.64 ± 6.85 a |
| | | C | 6.29 ± 1.14 bc | 28.08 ± 3.07 c | 10.39 ± 1.71 b |
| | | D | 14.92 ± 1.57 a | 51.67 ± 5.42 b | 25.54 ± 5.31 a |
| | male adults | A | 0.43 ± 0.15 b | 0.65 ± 0.12 c | 0.70 ± 0.18 c |
| | | B | 1.52 ± 0.41 ab | 15.95 ± 3.02 a | 5.10 ± 1.50 a |
| | | C | 0.98 ± 0.41 ab | 1.51 ± 0.31 c | 1.34 ± 0.29 bc |
| | | D | 2.50 ± 0.71 a | 4.53 ± 0.79 b | 3.70 ± 1.03 ab |

## 3.3. Population densities of *Bemisia tabaci* nymphs and adults in different positions on tomato plants under the different landscape types

The density of *B. tabaci* nymphs in the middle position of the tomato plants was significantly higher as compared to other positions, regardless of landscape types (flower landscape types in 2018: $F_{2,107} = 16.58$, $p < 0.001$; river landscape types in 2018: $F_{2,107} = 8.81$, $p < 0.001$; mountain landscape types in 2018: $F_{2,107} = 8.00$, $p = 0.001$; urban landscape types in 2018: $F_{2,116} = 23.50$, $p < 0.001$; flower landscape types in 2019: $F_{2,125} = 9.63$, $p < 0.001$; river landscape types in 2019: $F_{2,107} = 5.96$, $p = 0.004$; mountain landscape types in 2019: $F_{2,107} = 10.20$, $p < 0.001$; urban landscape types in 2019: $F_{2,107} = 18.03$, $p < 0.001$). Additionally, the density of adult *B. tabaci* in the upper position of the tomato plants was higher as compared to middle and lower positions of the tomato plants, regardless of the landscape types in both years (flower landscape types in 2018: $F_{2,107} = 182.67$, $p < 0.001$; river landscape types in 2018: $F_{2,107} = 34.70$, $p < 0.001$; mountain landscape types in 2018: $F_{2,107} = 34.54$, $p < 0.001$; urban landscape types in 2018: $F_{2,116} = 64.22$, $p < 0.001$; flower landscape types in 2019: $F_{2,125} = 98.40$, $p < 0.001$; river landscape types in 2019: $F_{2,107} = 25.90$, $p < 0.001$; mountain landscape types in 2019: $F_{2,107} = 50.43$, $p < 0.001$; urban landscape types in 2019: $F_{2,107} = 55.17$, $p < 0.001$) (figure 4).

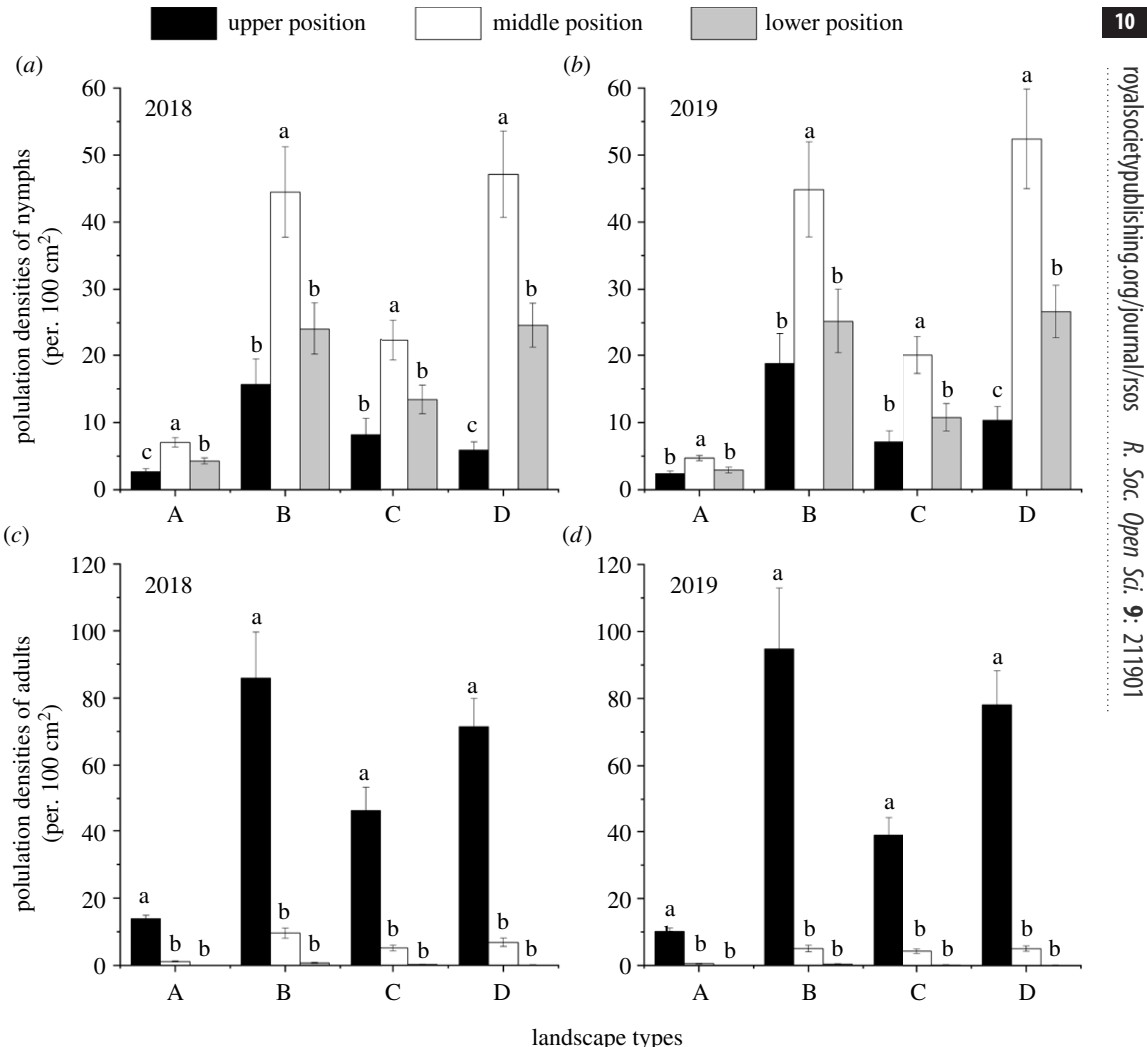

**Figure 4.** Population densities of *Bemisia tabaci* nymphs (*a,b*) and adults (*c,d*) (mean ± s.e.) in different positions of tomato plants under the different landscape types in Kunming, south China. Note: the different lowercase letters indicate a significant difference at the 0.05 level in the different landscape types during the same tomato planting year. A: landscape dominated by flower fields, B: landscape dominated by river, C: landscape dominated by mountain and D: landscape dominated by urban areas.

### 3.4. Yield of tomato under the different landscape types

The yields of tomato were different under different landscape types. The yields of tomato in the flower landscape type were more than 110 000 kg/ha in both years, which were the highest relative to those in the other landscape types, and the yields of tomato in 2018 in the river and the urban landscape types were the lowest relative to those in the other landscape types (in 2018: $F_{3,11} = 56.57$, $p < 0.001$; in 2019: $F_{3,11} = 11.19$, $p = 0.003$) (figure 5).

## 4. Discussion

### 4.1. Population dynamics of *Bemisia tabaci* in different landscape types

The densities of nymphs and female adults *B. tabaci* in the river landscape type were higher than those in the other landscape types. In this study, the purpose of setting up the river landscape type was to pay attention to the high humidity environment caused by the river. *B. tabaci* preferred a living environment with low humidity and dry, and the optimum relative humidity for nymph growth and development was 30%–70% RH [34]. An environment with low humidity and dry conditions is conducive to the activity of nymphs of *B. tabaci* [35]. Under high humidity conditions, the longevity

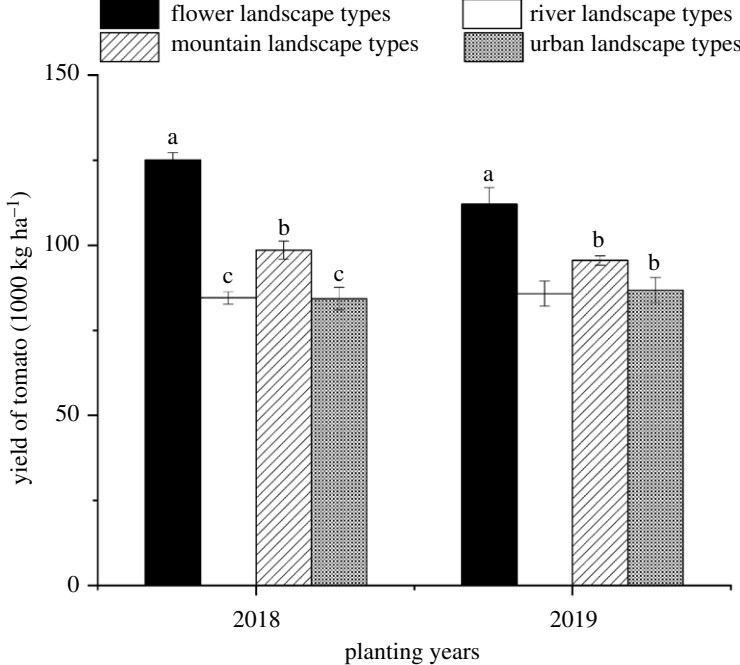

**Figure 5.** Yield of tomato (mean ± s.e.) under the different landscape types in Kunming, south China. Note: The different lowercase letters indicate a significant difference at the 0.05 level in the different landscape types during the same tomato planting year.

of adult *B. tabaci* was significantly shortened, and the number of eggs laid was reduced [36]. Therefore, the population densities of *B. tabaci* should have been restrained in the river landscape. However, there is an increasingly severe drought trend in Yunnan Province [37–39]. The annual rainfall in Kunming was only about 902.8 mm and 822.8 mm in 2018 and 2019, significantly, which was significantly lower than that in previous years [40,41]. During the whole sampling period, the average humidity in the tomato fields in the river landscape type was 45.28%. Li *et al*. pointed out that under 50% relative humidity, the development period of *B. tabaci* in tomato plants in Yunnan was the shortest as compared to other relative humidity, which was suitable for the growth and development of *B. tabaci* [42]. Therefore, the population density of *B. tabaci* in the river landscape type was higher than that in the other landscape types.

The population density of *B. tabaci* in the flower landscape type was lower than that in the other landscape types. Dou *et al*. pointed out that planting abundant flowering plants around the tomato field can effectively reduce the population density of *B. tabaci* in the tomato field, because flowering plants can enhance the control effect of natural enemies on *B. tabaci* [25]. Many studies have reported that flowering plants and abundant vegetation are beneficial to the diversity of natural enemies [43–45]. Flowering plants can provide food sources such as pollen and nectar for parasitoids and then improve the natural pest control ability of the environment [46]. Liu *et al*. determined that landscape diversity can enhance parasitism of cotton bollworm *Helicoverpa armigera* Hübner (Lepidoptera: Noctuidae) eggs by *Trichogramma chilonis* lshii (Hymenoptera: Trichogrammatidae) in cotton [28]. In our study, whether the diversity of natural enemies in the flower landscape affected the population of *B. tabaci* remains to be studied.

## 4.2. Population densities of *Bemisia tabaci* in different positions

In this study, adult *B. tabaci* were obviously concentrated in the upper position of tomato plants, and the nymphs were in the middle position of tomato plants in the tomato fields of all landscape types. *B. tabaci* is a phytophagous insect with piercing-sucking mouthparts. The adults like to feed on the young leaves in the upper position of the plant and lay eggs on these leaves [47,48]. Guo pointed out that adult *B. tabaci* tend to feed on leaves with high osmotic pressure at the top of cotton, tomato and other crops [49], which is consistent with the results of our study. After *B. tabaci* laid eggs on the upper young leaves of tomato, it took about one week for the eggs to develop into nymphs, and the whole nymph stage (1–4 instars) was about three weeks [50]. During this period, the tomato plants continued to grow, and the leaves in the

upper position upon which eggs were originally laid developed into middle leaves with the growth of plants. Therefore, in the main activity period of *B. tabaci* in tomato fields, integrated pest management should be focused on spraying the position of young leaves in the upper position of tomato plants to control the adult *B. tabaci*, and removal of the leaves with more nymphs should be done in the middle positions of tomato plants to control the *B. tabaci* nymphs.

The population dynamics and densities of nymphs and adults of *B. tabaci* in tomato fields in different landscape types were investigated in our study. However, *B. tabaci* is a complex of at least 36 biotypes [18,51]. The survival ability of different biotypes of *B. tabaci* was significantly different under different temperatures or environments, such as the number of eggs laid, the emergence rate and so on [52]. Therefore, the biotypes and the proportion of different biotypes of *B. tabaci* in tomato fields may be different in different landscape types or planting seasons. The *B. tabaci* samples collected in this study need to be further identified in order to clarify the influence of the different landscape types on the composition of biotypes of *B. tabaci*. This study only aimed to assess the impact of farmland landscape on *B. tabaci*. The agricultural ecosystem should be considered as a whole, as in addition to pests, parasitic, predatory natural enemies and entomogenous fungi also play important roles [53]. The next step of our study is to take these biotic factors into account and analyse them at the whole natural enemy sub-community level to explore the impact of different farmland landscapes on natural enemies. The original purpose of this study was to establish a green eco-agricultural landscape with sustainable development for many years. Therefore, we will continue this investigation for many years, analyse the possibility of green development of different types of landscape and truly establish a green eco-sustainable agricultural landscape.

Ethics. We declare that the work submitted for the publication is original and has not been published elsewhere; and that all the authors mutually agree with its content and have approved the paper for release and submission. All the authors have declared no conflict of interest. The manuscript does not contain experiments using animals and human studies. No permissions were required prior to conducting our fieldwork.

Data accessibility. All data used in these analyses is available in the electronic supplementary material [54].

Authors' contributions. S.Y.: formal analysis, investigation, methodology, writing—original draft and writing—review and editing; W.D.: data curation and investigation; M.L.: investigation; X.L.: investigation; Z.J.: investigation; G.C.: data curation, project administration, resources, supervision and writing—review and editing; X.Z.: conceptualization, data curation, funding acquisition, resources and writing—review and editing.

All authors gave final approval for publication and agreed to be held accountable for the work performed therein.

Conflict of interest declaration. We declare we have no competing interests.

Funding. The present research work was supported by the National Natural Science Foundation of China (grant no. 31760541), the Reserve Talent Project of Yunnan's Young and Middle-aged Academic and Technical Leaders (grant no. 202105AC160071), the Young Top Talents of 'High-level Talents Training Support Program in Yunnan Province' (grant no. YNWRQNBJ2020291) and the Reserve Talents Project for the 17th Batch of Kunming's Young and Middle-aged Academic and Technical Leaders (grant no. KMRCH2019023).

Acknowledgements. We thank Mr. Yu-Han Liu, Yuan Wang, Zi-Liao Wang, Fei Wang and Ms. Jian-Wen Lv, Xiao-Yun Wang (Yunnan Agricultural University of China) for helping us in sampling work. We also thank Mr. Jun-Yao Liu, Yong-Hua Yang, Yin-Long Wu and Ms. Feng-Ping Duan for helping us in field management.

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
