## [Peer Review File · Royal Society Open Science]

Review History

RSOS-210323.R0 (Original submission)

Review form: Reviewer 1

Is the manuscript scientifically sound in its present form?

No

Are the interpretations and conclusions justified by the results?

No

Is the language acceptable?

No

Do you have any ethical concerns with this paper?

No

Have you any concerns about statistical analyses in this paper?

No

Recommendation?

Reject

Comments to the Author(s)

In this study, the authors investigated the effects of four agricultural landscapes on the population dynamics of the pest *Bemisia tabaci* in tomato plantations in Yunnan province, China. The authors found out that the density of *Bemisia tabaci* adults and nymphs were higher in the “flower landscape type”. Overall, the manuscript has the potential, but several issues should be addressed first.

A careful linguist revision is needed. There are several grammatical mistakes throughout the text, most of which are merely disturbing, but on some occasions, they make the text unclear. The wording chosen is often verbose and could be dramatically simplified making the read easier. I have provided some suggestions at the end of the review, but these are not enough. Please, consider using the help of an English editor.

The introduction should improve both the structure and the rationale for the study. I think it would be better to start, as in the abstract, talking about the importance of agricultural landscapes for pests and pest control. This is a very large field of research nowadays and it is possible to talk in general terms before discussing the specific situation in China. The studied pest can be introduced after that. Why was it studied? Why in tomato fields? Did you have any hypotheses? Notice that currently, the purpose for this study is mentioned at L355-357, at the end of the discussion!

The material and methods section lacks important information and clarity. This part must enable the reader to repeat this experiment. Particularly, how were the landscape variables used in the PCA selected? which characteristic did you select? When was the sampling done? In which years and between which months? From the results, it is clear that the experiment was performed in 2018 and 2019, but this piece of information should not be a surprise for the reader. Also, clarify that you recorded both nymphs and adults.

Moreover, there are not enough details about the study sites. Just to say that the “river landscape” was the one where the main elements are rivers does not tell much. I would suggest writing a small sub-paragraph for each of the landscapes describing their main characteristics. It seems to me that the authors used a ‘lazy’ description, perhaps even a circular argument, to define the landscapes. For instance, L151-152, “Three of them are divided into urban landscape type, their main landscape elements are urban”.

Figure 1 is OK, but if the authors could add photographs of the study sites that could also be useful.

Figure 2 is unclear. Which landscape variables were considered for the PCA? This is not specified in the text.

Finally, the statistical analyses are not very clear. ANOVA was used for the census data. But what were the independent factors?

The results section can also be improved. Perhaps the interpretation of the PCA, currently the first paragraph of results, should go in the methods because this is what you used to discriminate between landscapes. In paragraph 3.1, the authors provide the actual value for the eigenvectors, which are essentially distance to create the PCA plot. Is there any reason to present those number? As far I know, the number in itself is not important. If so, please justify this in the text. The results of paragraph 3.4 would be easier to interpret with the help of one figure where you compared densities on the different parts of the plant in the four landscapes and years instead of table 3. The same can be said about the results of paragraph 3.5.

In the discussion, the reader realises that one of the reasons why the river landscape was selected is that the authors expected the higher humidity levels to be unfavourable for *Bemisia tabaci*. I think that this is reasonable information that should be used before to explain why those landscapes were studied and what the authors expected to find out.

Also, be careful at stating that the high abundance of natural enemies in the flower landscape (L323) can explain your results because you haven't measured natural enemies' abundance. Additionally, I feel there is a contradiction when you say that you search for green, eco-sustainable landscapes, but you tested on which part of the plants *B. tabaci* densities were higher to know where to spray pesticides. I would imagine that justifying the rationale for this study in the introduction will also help in this part.

More generally, you don't compare your results with many other studies. If there is no previous data on *B. tabaci* densities in this crop or from the region, you should state it.

The abstract requires a careful linguist revision. I have included some comments, but various sentences are unclear and need to be rephrased. The authors mention four landscapes: 1) flower fields, 2) mountain, 3) river, and 4) urban, but should specify how those were characterised and perhaps briefly why they were selected. Meadows can occur at different elevations, also on mountains. Mountain and river landscapes do not tell much about what habitats are present in the area.

Two of the keywords, '*Bemisia tabaci*' and 'Agricultural landscape', are already included in the title and the authors could think about something better.

Reference 23, Gabor is a given name. Reference 30, *Anisodactylus signatus* should be in italics.

Specific comments:

L11 I would remove "and stability". Changes and stability in a population is already part of the concept of population dynamics. L12 change "four typical landscape types" to "four landscapes."

L14, ff. change "tomato planting fields" to "tomato fields"

L15-20. There are problems with this long sentence. At L17 the verb is missing. At L19 the flower fields landscape is mentioned but providing two different density values and a "respectively", which suggest that another landscape should have been also mentioned in the sentence. Or those were the densities of nymphs and adults? Please rephrase.

L22 if you say "were the highest" you can remove "than that in the other landscape types".

L23 again there are two density values and a respectively but when referring only to a single habitat type (flower landscapes).

L24 the middle position of the plant? or what? this is not specified.

L39-40 I would remove this sentence because the agricultural landscapes are variable and much depends on the scale you consider. I think that it is reasonable to start the paper with the problem of agricultural intensification that you have presented well at L40-43. You could also consider talking more generally, because what it's true for China holds, unfortunately, also for many other countries. Perhaps later you can get more into the specific situations of China.

L43 in agroecosystems

L44 change "will" to "can"

L45 is "population stability" necessary? I think this is redundant with the concept of population dynamics.

L50 which supports pest control.

L54-56 The justification of the rationale is weak. Under which scenarios pests and more favoured than natural enemies? Which environmental factors did you refer to? Why was the study done in tomato plantations and not in another crop?

L74-75 unclear, please rephrase. Which landscape variables were considered?

L89 both years? which years?

L94 this sentence does not make much sense unless you specify how was the sampling done instead of saying "by sampling".

L123 "Four spikes of tomato were reserved in the tomato plants in the fields." What was the purpose of this? Perhaps there is nothing wrong with it, but could you check the expression "spike of tomato"?

L126 to be weighed

L131 were compared

L132 densities of different positions? On the plant you mean? Unclear.

L135 change "made by" to "create using"

L140 accounted for

L146 rivers

L149 mountains

L151-152 Three of them are divided into urban landscape type, their main landscape elements are urban. What does it mean that they are "urban"?

L158 was longer than

L159 Aug. if you want to be consistent, but I think later on it should be Sept.

L164-165 "The main activity period of *B. tabaci* nymphs in the same period" is confusing. Could you please rephrase?

L203-204, 227-228 "regardless of the planting years" means "in both years"?

L290-292 Please revise the following sentence: "The yield of tomato in the tomato planting fields in the flower landscape types, more than 110 000 kg / ha". Avoid unnecessary repetition, if you say 'tomato yield', clearly, it is from the tomato fields. Do not forget the verb.

L327 remove "kind of"

Review form: Reviewer 2

Is the manuscript scientifically sound in its present form?

Yes

Are the interpretations and conclusions justified by the results?

Yes

Is the language acceptable?

Yes

Do you have any ethical concerns with this paper?

No

Have you any concerns about statistical analyses in this paper?

No

Recommendation?

Accept with minor revision (please list in comments)

Comments to the Author(s)

I recommend this MS for publication with minor corrections as given below:

L1-3 title may be revised as "Regulation of dynamics and densities of Whitefly *Bemisia tabaci* by agricultural landscapes in south China"

L12 We selected four typical landscape types (flower field, mountain, river, and urban) as the main elements in Yunnan province, South China.

L35 polyphagous in nature, remove species, remove comma after hosts
 L36 remove comma after cotton
 L36 The nymphs and adults of *B. tabaci* can cause
 L37 economic damage directly by sucking plant fluids and indirectly by transmitting plant viruses including
 L39-40 urban area,
 L44 The change in agricultural landscape pattern
 L67 conducive in decreasing
 L89-90 please mentioned the years of study
 L91 against insect or disease pests in our experiments
 L97 Leaves with *B. tabaci* nymphs and adults
 L98 using the fixed five points sampling method
 101 remove coma (,) before and
 L102 observed per plot
 L103 remove coma (,) before and
 L112 remove coma (,) after main
 L112 and end for each of these,
 L113-118 must be in past sentences
 L134 $P=0.05$
 L135 figures of population dynamics and cumulative seasonal activity curves
 L167 write full September
 L177 it is better to use "as compared to" instead of "than that in the" in whole results sections
 L266 better use 146.88 for F and 0.001 for P

Decision letter (RSOS-210323.R0)

Dear Mr Yang

The Editors assigned to your paper RSOS-210323 "Regulation of densities and dynamics of Whitefly *Bemisia tabaci* by agricultural landscapes in south China" have made a decision based on their reading of the paper and any comments received from reviewers.

Regrettably, in view of the reports received, the manuscript has been rejected in its current form. However, a new manuscript may be submitted which takes into consideration these comments.

We invite you to respond to the comments supplied below and prepare a resubmission of your manuscript. Below the referees' and Editors' comments (where applicable) we provide additional requirements. We provide guidance below to help you prepare your revision.

Please note that resubmitting your manuscript does not guarantee eventual acceptance, and we do not generally allow multiple rounds of revision and resubmission, so we urge you to make every effort to fully address all of the comments at this stage. If deemed necessary by the Editors, your manuscript will be sent back to one or more of the original reviewers for assessment. If the original reviewers are not available, we may invite new reviewers.

Please resubmit your revised manuscript and required files (see below) no later than 02-Feb-2022. Note: the ScholarOne system will 'lock' if resubmission is attempted on or after this deadline. If

you do not think you will be able to meet this deadline, please contact the editorial office immediately.

Please note article processing charges apply to papers accepted for publication in Royal Society Open Science (<https://royalsocietypublishing.org/rsos/charges>). Charges will also apply to papers transferred to the journal from other Royal Society Publishing journals, as well as papers submitted as part of our collaboration with the Royal Society of Chemistry (<https://royalsocietypublishing.org/rsos/chemistry>). Fee waivers are available but must be requested when you submit your manuscript (<https://royalsocietypublishing.org/rsos/waivers>).

Thank you for submitting your manuscript to Royal Society Open Science and we look forward to receiving your resubmission. If you have any questions at all, please do not hesitate to get in touch.

on behalf of Dr Polly Campbell (Associate Editor) and Pete Smith (Subject Editor)
openscience@royalsociety.org

Associate Editor Comments to Author (Dr Polly Campbell):

First, I want to apologize to the authors for the long delay in getting a decision on their manuscript. The main result of this study (i.e. the dramatic reduction in *B. tabaci* in one landscape relative to all others) is interesting and potentially valuable. However, the value of this result relies on accurate and informative characterization of landscape types. Defining landscapes by topographical features like mountains and rivers is insufficient and the relevance of these descriptors to a study of pest abundance is unclear. Reviewer 1 provides a constructive critique of these and other aspects of the manuscript that leave room for improvement. This reviewer's suggestions, together with specific edits suggested by both reviewers, should help the authors in the revision process.

A couple of specific comments:

L48-50 please clarify "natural enemies". Are these predators of pest species? Competitors?

L57 Please provide a citation to support this statement.

Fig 2 I could not extract anything meaningful from this figure as presented, nor could I connect it to the landscape patterns described in L83-88 and 139-154 (why is this in both methods and results?). If this analysis is retained, please find a more informative way to present this result and please make sure that the verbal description relates to what is shown in the figure.

Please reference the supplementary tables/data in the main text. For example, the excel table currently identified as Covariance matrix of PCA should be numbered and referenced in the section dealing with these data.

Reviewer comments to Author:

Reviewer: 1

Comments to the Author(s)

In this study, the authors investigated the effects of four agricultural landscapes on the population dynamics of the pest *Bemisia tabaci* in tomato plantations in Yunnan province, China. The authors found out that the density of *Bemisia tabaci* adults and nymphs were higher in the “flower landscape type”. Overall, the manuscript has the potential, but several issues should be addressed first.

A careful linguist revision is needed. There are several grammatical mistakes throughout the text, most of which are merely disturbing, but on some occasions, they make the text unclear. The wording chosen is often verbose and could be dramatically simplified making the read easier. I have provided some suggestions at the end of the review, but these are not enough. Please, consider using the help of an English editor.

The introduction should improve both the structure and the rationale for the study. I think it would be better to start, as in the abstract, talking about the importance of agricultural landscapes for pests and pest control. This is a very large field of research nowadays and it is possible to talk in general terms before discussing the specific situation in China. The studied pest can be introduced after that. Why was it studied? Why in tomato fields? Did you have any hypotheses? Notice that currently, the purpose for this study is mentioned at L355-357, at the end of the discussion!

The material and methods section lacks important information and clarity. This part must enable the reader to repeat this experiment. Particularly, how were the landscape variables used in the PCA selected? which characteristic did you select? When was the sampling done? In which years and between which months? From the results, it is clear that the experiment was performed in 2018 and 2019, but this piece of information should not be a surprise for the reader. Also, clarify that you recorded both nymphs and adults.

Moreover, there are not enough details about the study sites. Just to say that the “river landscape” was the one where the main elements are rivers does not tell much. I would suggest writing a small sub-paragraph for each of the landscapes describing their main characteristics. It seems to me that the authors used a ‘lazy’ description, perhaps even a circular argument, to define the landscapes. For instance, L151-152, “Three of them are divided into urban landscape type, their main landscape elements are urban”.

Figure 1 is OK, but if the authors could add photographs of the study sites that could also be useful.

Figure 2 is unclear. Which landscape variables were considered for the PCA? This is not specified in the text.

Finally, the statistical analyses are not very clear. ANOVA was used for the census data. But what were the independent factors?

The results section can also be improved. Perhaps the interpretation of the PCA, currently the first paragraph of results, should go in the methods because this is what you used to discriminate between landscapes. In paragraph 3.1, the authors provide the actual value for the eigenvectors, which are essentially distance to create the PCA plot. Is there any reason to present those number? As far I know, the number in itself is not important. If so, please justify this in the text. The results of paragraph 3.4 would be easier to interpret with the help of one figure where you compared densities on the different parts of the plant in the four landscapes and years instead of table 3. The same can be said about the results of paragraph 3.5.

In the discussion, the reader realises that one of the reasons why the river landscape was selected is that the authors expected the higher humidity levels to be unfavourable for *Bemisia tabaci*. I

think that this is reasonable information that should be used before to explain why those landscapes were studied and what the authors expected to find out.

Also, be careful at stating that the high abundance of natural enemies in the flower landscape (L323) can explain your results because you haven't measured natural enemies' abundance.

Additionally, I feel there is a contradiction when you say that you search for green, eco-sustainable landscapes, but you tested on which part of the plants *B. tabaci* densities were higher to know where to spray pesticides. I would imagine that justifying the rationale for this study in the introduction will also help in this part.

More generally, you don't compare your results with many other studies. If there is no previous data on *B. tabaci* densities in this crop or from the region, you should state it.

The abstract requires a careful linguist revision. I have included some comments, but various sentences are unclear and need to be rephrased. The authors mention four landscapes: 1) flower fields, 2) mountain, 3) river, and 4) urban, but should specify how those were characterised and perhaps briefly why they were selected. Meadows can occur at different elevations, also on mountains. Mountain and river landscapes do not tell much about what habitats are present in the area.

Two of the keywords, '*Bemisia tabaci*' and 'Agricultural landscape', are already included in the title and the authors could think about something better.

Reference 23, Gabor is a given name. Reference 30, *Anisodactylus signatus* should be in italics.

Specific comments:

L11 I would remove "and stability". Changes and stability in a population is already part of the concept of population dynamics. L12 change "four typical landscape types" to "four landscapes:"

L14, ff. change "tomato planting fields" to "tomato fields"

L15-20. There are problems with this long sentence. At L17 the verb is missing. At L19 the flower fields landscape is mentioned but providing two different density values and a "respectively", which suggest that another landscape should have been also mentioned in the sentence. Or those were the densities of nymphs and adults? Please rephrase.

L22 if you say "were the highest" you can remove "than that in the other landscape types".

L23 again there are two density values and a respectively but when referring only to a single habitat type (flower landscapes).

L24 the middle position of the plant? or what? this is not specified.

L39-40 I would remove this sentence because the agricultural landscapes are variable and much depends on the scale you consider. I think that it is reasonable to start the paper with the problem of agricultural intensification that you have presented well at L40-43. You could also consider talking more generally, because what it's true for China holds, unfortunately, also for many other countries. Perhaps later you can get more into the specific situations of China.

L43 in agroecosystems

L44 change "will" to "can"

L45 is "population stability" necessary? I think this is redundant with the concept of population dynamics.

L50 which supports pest control.

L54-56 The justification of the rationale is weak. Under which scenarios pests and more favoured than natural enemies? Which environmental factors did you refer to? Why was the study done in tomato plantations and not in another crop?

L74-75 unclear, please rephrase. Which landscape variables were considered?

L89 both years? which years?

L94 this sentence does not make much sense unless you specify how was the sampling done instead of saying "by sampling".

L123 "Four spikes of tomato were reserved in the tomato plants in the fields." What was the purpose of this? Perhaps there is nothing wrong with it, but could you check the expression "spike of tomato"?

L126 to be weighed

L131 were compared

L132 densities of different positions? On the plant you mean? Unclear.

L135 change "made by" to "create using"

L140 accounted for

L146 rivers

L149 mountains

L151-152 Three of them are divided into urban landscape type, their main landscape elements are urban. What does it mean that they are "urban"?

L158 was longer than

L159 Aug. if you want to be consistent, but I think later on it should be Sept.

L164-165 "The main activity period of *B. tabaci* nymphs in the same period" is confusing. Could you please rephrase?

L203-204, 227-228 "regardless of the planting years" means "in both years"?

L290-292 Please revise the following sentence: "The yield of tomato in the tomato planting fields in the flower landscape types, more than 110 000 kg / ha". Avoid unnecessary repetition, if you say 'tomato yield', clearly, it is from the tomato fields. Do not forget the verb.

L327 remove "kind of"

Reviewer: 2

Comments to the Author(s)

I recommend this MS for publication with minor corrections as given below:

L1-3 title may be revised as "Regulation of dynamics and densities of Whitefly *Bemisia tabaci* by agricultural landscapes in south China"

L12 We selected four typical landscape types (flower field, mountain, river, and urban) as the main elements in Yunnan province, South China.

L35 polyphagous in nature, remove species, remove comma after hosts

L36 remove comma after cotton

L36 The nymphs and adults of *B. tabaci* can cause

L37 economic damage directly by sucking plant fluids and indirectly by transmitting plant viruses including

L39-40 urban area,

L44 The change in agricultural landscape pattern

L67 conducive in decreasing

L89-90 please mentioned the years of study

L91 against insect or disease pests in our experiments

L97 Leaves with *B. tabaci* nymphs and adults

L98 using the fixed five points sampling method

101 remove coma (,) before and

L102 observed per plot

L103 remove coma (,) before and

L112 remove coma (,) after main

L112 and end for each of these,

L113-118 must be in past sentences

L134 $P=0.05$

L135 figures of population dynamics and cumulative seasonal activity curves

L167 write full September

L177 it is better to use “as compared to” instead of “than that in the” in whole results sections
 L266 better use 146.88 for F and 0.001 for P

===PREPARING YOUR MANUSCRIPT===

===PREPARING YOUR REVISION IN SCHOLARONE===

<https://royalsociety.org/journals/authors/author-guidelines/#supplementary-material> to include a suitable title and informative caption. An example of appropriate titling and captioning may be found at https://figshare.com/articles/Table_S2_from_Is_there_a_trade-off_between_peak_performance_and_performance_breadth_across_temperatures_for_aerobic_sc_ope_in_teleost_fishes_/3843624.

Author's Response to Decision Letter for (RSOS-210323.R0)

See Appendix A.

RSOS-210323.R1 (Revision)

Review form: Reviewer 3

Is the manuscript scientifically sound in its present form?

Yes

Are the interpretations and conclusions justified by the results?

Yes

Is the language acceptable?

Yes

Do you have any ethical concerns with this paper?

No

Have you any concerns about statistical analyses in this paper?

No

Recommendation?

Accept with minor revision (please list in comments)

Comments to the Author(s)

The comments to authors:

The manuscript addressed RSOS-211901 of Royal Society Open Science reported the dynamics and densities of *Bemisia tabaci* in different agricultural landscapes in south China. Overall, the MS is well written, included relevant references, and the statistical analyses are OK. I recommend this MS for publication with minor corrections as given below:

Line 1 is Whitefly, but the correct one is whitefly

Line 13 the correct one is Yunnan Province, south China

Line 16 is density, but the correct one is densities

Line 41-43 These diverse noncrop habitats can provide foods, shelters, alternative hosts, and favorable microclimates for natural enemies in the field, which supports pest control.

Line 51-52 add Abd-Rabou S and Simmons AM. Survey of reproductive host plants of *Bemisia tabaci* (Hemiptera: Aleyrodidae) in Egypt, including new host records [J]. Entomological News, 2010, 121(5): 456-465.

Line 60, 62, 67, 314 Province

Line 65 delete comma (,) after river

Line 78 south China

Line 96 is main types, but the correct one is main species

Line 99 is rivers, but the correct one is river

Line 135 delete comma (,) after 50

Line 156 delete comma (,) after main

Line 166-168 In 2018, the main activity period of *B. tabaci* nymphs in the flower landscape type, as long as 52 days, was longer as compared to other landscape types. The peak activity date in the tomato fields in the river landscape type, 28 Aug.,.....

Line 170 were later relative to.....

Line 173 main activity periods

Line 175 were in early September.

Line 181-182, 192, 196, 202 in the flower landscape type

Line 183, 226, 261 in the urban landscape type

Line 189 and their peak activity date were in mid-Sept..

Line 198, 253, 264, 304, 321 in the river landscape type

Line 219 In the mountain landscape type,

Line 255 delete (, were the highest relative to the other landscape types)

Line 255-256 At each activity period

Line 271 At the early and late activity period, the densities of male adult *B. tabaci* were.....

Line 273 At the main activity period

Line 279-281 The density of *B. tabaci* nymphs in the middle position of the tomato plants was significantly higher as compared to other positions, regardless of landscape types

Line 286-287 Additionally, the density of adult *B. tabaci* in the upper position of the tomato plants was higher as.....

Line 296-298 The yield of tomato were different under different landscape types. The yield of tomato in the flower landscape type were more than 110 000 kg / ha in both years, which were the highest.....

Line 331 After the scientific names mentioned for the first time, the names of author, order and family must be placed.

Line 336 in the middle position

Line 463 Lövei GL

Decision letter (RSOS-211901.R1)

Dear Mr Yang

On behalf of the Editors, we are pleased to inform you that your Manuscript RSOS-211901 "Regulation of dynamics and densities of Whitefly *Bemisia tabaci* by agricultural landscapes in south China" has been accepted for publication in Royal Society Open Science subject to minor revision in accordance with the referees' reports. Please find the referees' comments along with any feedback from the Editors below my signature.

Please submit your revised manuscript and required files (see below) no later than 7 days from today's (ie 21-Feb-2022) date. Note: the ScholarOne system will 'lock' if submission of the revision is attempted 7 or more days after the deadline. If you do not think you will be able to meet this deadline please contact the editorial office immediately.

on behalf of Dr Polly Campbell (Associate Editor) and Pete Smith (Subject Editor)
 openscience@royalsociety.org

Associate Editor Comments to Author (Dr Polly Campbell):

Comments to the Author:

The revised version of this manuscript is greatly improved and the authors' careful attention to detail is appreciated. Please address all the reviewer's minor comments before final submission.

Reviewer comments to Author:

Reviewer: 3

Comments to the Author(s)

The comments to authors:

The manuscript addressed RSOS-211901 of Royal Society Open Science reported the dynamics and densities of *Bemisia tabaci* in different agricultural landscapes in south China. Overall, the MS is well written, included relevant references, and the statistical analyses are OK. I recommend this MS for publication with minor corrections as given below:

Line 1 is Whitefly, but the correct one is whitefly

Line 13 the correct one is Yunnan Province, south China

Line 16 is density, but the correct one is densities

Line 41-43 These diverse noncrop habitats can provide foods, shelters, alternative hosts, and favorable microclimates for natural enemies in the field, which supports pest control.

Line 51-52 add Abd-Rabou S and Simmons AM. Survey of reproductive host plants of *Bemisia tabaci* (Hemiptera: Aleyrodidae) in Egypt, including new host records [J]. *Entomological News*, 2010, 121(5): 456-465.

Line 60, 62, 67, 314 Province

Line 65 delete comma (,) after river

Line 78 south China

Line 96 is main types, but the correct one is main species

Line 99 is rivers, but the correct one is river

Line 135 delete comma (,) after 50

Line 156 delete comma (,) after main

Line 166-168 In 2018, the main activity period of *B. tabaci* nymphs in the flower landscape type, as long as 52 days, was longer as compared to other landscape types. The peak activity date in the tomato fields in the river landscape type, 28 Aug.,.....

Line 170 were later relative to.....

Line 173 main activity periods

Line 175 were in early September.

Line 181-182, 192, 196, 202 in the flower landscape type

Line 183, 226, 261 in the urban landscape type

Line 189 and their peak activity date were in mid-Sept..

Line 198, 253, 264, 304, 321 in the river landscape type

Line 219 In the mountain landscape type,

Line 255 delete (, were the highest relative to the other landscape types)

Line 255-256 At each activity period

Line 271 At the early and late activity period, the densities of male adult *B. tabaci* were.....

Line 273 At the main activity period

Line 279-281 The density of *B. tabaci* nymphs in the middle position of the tomato plants was significantly higher as compared to other positions, regardless of landscape types

Line 286-287 Additionally, the density of adult *B. tabaci* in the upper position of the tomato plants was higher as.....

Line 296-298 The yield of tomato were different under different landscape types. The yield of tomato in the flower landscape type were more than 110 000 kg / ha in both years, which were the highest.....

Line 331 After the scientific names mentioned for the first time, the names of author, order and family must be placed.

Line 336 in the middle position

Line 463 Lövei GL

===PREPARING YOUR MANUSCRIPT===

one version should clearly identify all the changes that have been made (for instance, in coloured highlight, in bold text, or tracked changes);

===PREPARING YOUR REVISION IN SCHOLARONE===

-- If you are requesting an article processing charge waiver, you must select the relevant waiver option (if requesting a discretionary waiver, the form should have been uploaded, see 'File upload' above).

-- If you have uploaded any electronic supplementary (ESM) files, please ensure you follow the guidance at <https://royalsociety.org/journals/authors/author-guidelines/#supplementary-material> to include a suitable title and informative caption. An example of appropriate titling and captioning may be found at https://figshare.com/articles/Table_S2_from_Is_there_a_trade-off_between_peak_performance_and_performance_breadth_across_temperatures_for_aerobic_scope_in_teleost_fishes_/3843624.

Author's Response to Decision Letter for (RSOS-211901.R1)

See Appendix B.

Decision letter (RSOS-211901.R2)

Dear Mr Yang,

I am pleased to inform you that your manuscript entitled "Regulation of dynamics and densities of whitefly *Bemisia tabaci* by agricultural landscapes in south China" is now accepted for publication in Royal Society Open Science.

on behalf of Dr Polly Campbell (Associate Editor) and Pete Smith (Subject Editor)
openscience@royalsociety.org

Appendix A

Dr Polly Campbell
Associate Editor, Royal Society Open Science

03 Dec. 2021

Dear Dr Polly Campbell,

We are receipt of the set of comments by reviewers, and we thank you for the efficient handling of our manuscript. We are happy to read that the editor and reviewers found our study (RSOS-210323) interesting and potentially valuable. They provided valuable suggestions, each of which we have carefully considered. We added our replies after each comment separately – please find these below.

As a result, we revised our manuscript that we hope it is now easier to be read and comprehended. We hope you will find our revision satisfactory, and our MS can be accepted for publication in Royal Society Open Science.

Again, thank you for your editorial assistance, and we await further from you concerning our manuscript.

Sincerely yours

Dr. Xiaoming Zhang

Corresponding author

Associate Editor Comments

Point 1: L48-50 please clarify “natural enemies”. Are these predators of pest species? Competitors?

Response: We agree with the comment, and we have clarified natural enemies are parasitoids and predators. Please see line 67 in revision version.

Point 2: L57 Please provide a citation to support this statement.

Response: We agree with the comment. We have provided two references to support this statement that Yunnan province is one of the regions with the richest biodiversity in the world. Please see lines 86-87 and 652-657 in revision version.

Point 3: Fig 2 I could not extract anything meaningful from this figure as presented, nor could I connect it to the landscape patterns described in L83-88 and 139-154 (why is this in both methods and results?). If this analysis is retained, please find a more informative way to present this result and please make sure that the verbal description relates to what is shown in the figure.

Response: We agree with the comment. Combined with the comment from Reviewer 1 (Point 10), we have deleted Figure 2 and upload the mapping data as supplementary data 1 (Data S1). Please see lines 122 and 808-809 in revision version.

Point 4: Please reference the supplementary tables/data in the main text. For example, the excel table currently identified as Covariance matrix of PCA should be numbered and referenced in the section dealing with these data.

Response: We agree with the comment. We have numbered and referenced the supplementary data (Covariance matrix of PCA) in the main text. Please see line 122 in revision version.

Reviewer: 1

Point 5: A careful linguist revision is needed. There are several grammatical mistakes throughout the text, most of which are merely disturbing, but on some occasions, they make the text unclear. The wording chosen is often verbose and could be dramatically simplified making the read easier. I have provided some suggestions at the end of the review, but these are not enough. Please, consider using the help of an English editor.

Response: We agree with the comment. To make our manuscript clear, correct, and unambiguous, after we revised our manuscript, we invited DBMediting for professional English language editing services. The English service company has professional English editing qualification, and the English papers edited by this company have been published more than 100 in major journals.

Point 6: The introduction should improve both the structure and the rationale for the study. I think it would be better to start, as in the abstract, talking about the importance of agricultural landscapes for pests and pest control. This is a very large field of research nowadays and it is possible to talk in general terms before discussing the specific situation in China. The studied pest can be introduced after that. Why was it studied? Why in tomato fields? Did you have any hypotheses? Notice that currently, the purpose for this study is mentioned at L355-357, at the end of the discussion!

Response: We agree with the comment. We have improved the introduction both the structure and the rationale. Firstly, we talked about the importance of agricultural landscapes for pests and pest control; secondly, we introduced the studied pest *Bemisia tabaci*; thirdly, we explained why we studied this pest, and we have present hypotheses in the third paragraph of the introduction. And in the last paragraph, we mentioned the purpose for our study. Please see lines 59-98 in revision version.

Point 7: The material and methods section lacks important information and clarity. This part must enable the reader to repeat this experiment. Particularly, how were the landscape variables used in the PCA selected? which characteristic did you select? When was the sampling done? In which years and between which months? From the results, it is clear that the experiment was performed in 2018 and 2019, but this piece of information should not be a surprise for the reader. Also, clarify that you recorded both nymphs and adults.

Response: We agree with the comments.

① We have added some information in the material and methods, including the description of different landscapes. To determine the land cover types, it was selected by use of Google Earth Profession and field inspections (ground-truthing) once a month during the tomato growing seasons in 2018 and 2019. The cover types in each landscape were divided into 10 types according to vegetation type, human factor interference and land type characteristics. Please see lines 104-136 in revision version.

② In each 800 m² plot (field plots: 20 m×40 m; plant spacing: 40 cm; row spacing: 80 cm) of the tomato fields in each landscape, the first sampling started 10 days after tomato transplanting in 2018 and 2019. Leaves with *B. tabaci* nymphs and adults were collected every 10 days using the fixed five points sampling method until the end of the tomato growing season. Please see lines 148-151 in revision version.

Point 8: Moreover, there are not enough details about the study sites. Just to say that the “river landscape” was the one where the main elements are rivers does not tell much. I would suggest writing a small sub-paragraph for each of the landscapes describing their main characteristics. It seems to me that the authors used a ‘lazy’ description, perhaps even a circular argument, to define the landscapes. For instance, L151-152, “Three of them are divided into urban landscape type, their main landscape elements are urban”.

Response: We agree with the comment. We have written small sub-paragraph for each of the landscapes describing their main characteristics. Please see lines 123-136 in revision version.

Point 9: Figure 1 is OK, but if the authors could add photographs of the study sites that could also be useful.

Response: We agree with the comment. We have added photographs of the study sites in Figure 1. Please see lines 804-807 in revision version.

Point 10: Figure 2 is unclear. Which landscape variables were considered for the PCA? This is not specified in the text.

Response: We agree with the comments.

① Combined with the comment from Associate Editor (**Point 3**), we have deleted Figure 2 and upload the mapping data as supplementary data 1 (Data S1). Please see lines 122 and 808-809 in revision version.

② To determine the land cover types, it was selected by use of Google Earth Profession and field inspections (ground-truthing) once a month during the tomato growing seasons in 2018 and 2019. The cover types in each landscape were divided into 10 types according to vegetation type, human factor interference and land type characteristics. Please see lines 104-118 in revision version.

Point 11: Finally, the statistical analyses are not very clear. ANOVA was used for the census data. But what were the independent factors?

Response: We agree with the comment, and we have modified this part. Census data were initially subjected to a one-way ANOVA (repeated measures) after tests of normality (Shapiro–Wilk) and homoscedasticity (Bartlett), with agriculture landscape types as the main effect. To reduce the impact of occurrence time on the population densities of *B. tabaci*, the activity period of *B. tabaci* was divided into early, main, and late activity period by quartile method. Differences in *B. tabaci* densities were compared among agriculture landscape types in the same activity period, as well as densities in different positions of the tomato plants, constrained by landscape type (identified as above) by Least Significant Difference ($P=0.05$). Please see lines 186-195 in revision version.

Point 12: The results section can also be improved. Perhaps the interpretation of the PCA, currently the first paragraph of results, should go in the methods because this is what you used to discriminate between landscapes. In paragraph 3.1, the authors provide the actual value for the eigenvectors, which are essentially distance to create the PCA plot. Is there any reason to present those number? As far I know, the number in itself is not important. If so, please justify this in the text. The results of paragraph 3.4 would be easier to interpret with the help of one figure where you compared densities on the different parts of the plant in the four landscapes and years instead of table 3. The same can be said about the results of paragraph 3.5.

Response: We agree with the comments.

① We have removed the interpretation of the PCA in the first paragraph of results, and we

have supplemented interpretation of the PCA in the methods and removed the actual value for the eigenvectors. Please see lines 121-136 in revision version.

② We have removed the table 3 and 4, and we use two figures to interpret the results of those paragraphs. Please see lines 761-771 and 814-817 in revision version.

Point 13: In the discussion, the reader realises that one of the reasons why the river landscape was selected is that the authors expected the higher humidity levels to be unfavourable for *Bemisia tabaci*. I think that this is reasonable information that should be used before to explain why those landscapes were studied and what the authors expected to find out.

Response: We agree with the comment. We have supplemented the reasons why we chose these landscape types in paragraph 2.1. Please see lines 127-130 in revision version.

Point 14: Also, be careful at stating that the high abundance of natural enemies in the flower landscape (L323) can explain your results because you haven't measured natural enemies' abundance.

Response: We agree with the comment. We have cited literature to explain high abundance of natural enemies in the flower landscape. Please see lines 385-388 in revision version.

Point 15: Additionally, I feel there is a contradiction when you say that you search for green, eco-sustainable landscapes, but you tested on which part of the plants *B. tabaci* densities were higher to know where to spray pesticides. I would imagine that justifying the rationale for this study in the introduction will also help in this part.

Response: We agree with the comment. We have rephrased this part to make it clearer. Please see lines 97-98 and 412-413 in revision version.

Point 16: More generally, you don't compare your results with many other studies. If there is no previous data on *B. tabaci* densities in this crop or from the region, you should state it.

Response: We agree with the comment. We have cited literature to compare our results with many other studies. Please see lines 378-381 and 385-388 in revision version.

Point 17: The abstract requires a careful linguist revision. I have included some comments, but various sentences are unclear and need to be rephrased. The authors mention four landscapes: 1) flower fields, 2) mountain, 3) river, and 4) urban, but should specify how those were characterised and perhaps briefly why they were selected. Meadows can occur at different elevations, also on mountains. Mountain and river landscapes do not tell much about what habitats are present in the area.

Response: We agree with the comments.

① We have rephrased relevant description in abstract, please see lines 11-30 in revision version.

② In the mountain landscape types, their main landscape cover types were cypress forest with altitude difference more than 150 m and there was no overlap with other landscape cover types. The purpose of setting up the river landscape type was to pay attention to the high humidity environment caused by the river. We have divided landscape cover types such as vegetable fields, orchards or wasteland around the river, and the river landscape cover types only includes the river in our manuscript. We have rephrased relevant description in our manuscript, please see lines 106-118 in revision version.

Point 18: Two of the keywords, '*Bemisia tabaci*' and 'Agricultural landscape', are already included in the title and the authors could think about something better.

Response: We agree with the comment, and we have chosen "Population dynamic", "Population density", "Principal Components Analysis", and "Quartile method" as the keywords of our manuscript. Please see lines 31-32 in revision version.

Point 19: Reference 23, Gabor is a given name. Reference 30, *Anisodactylus signatus* should be in italics.

Response: We agree with the comment. We have revised the name "Gabor LL" to "Lövei GL". And we have revised "*Anisodactylus signatus*" to "*Anisodactylus signatus*". Please see lines 658 and 684 in revision version.

Specific comments:

Point 20: L11 I would remove "and stability". Changes and stability in a population is already part of the concept of population dynamics. L12 change "four typical landscape types" to "four landscapes:"

Response: We agree with the comment. We have removed "and stability". And combined with the comment from Reviewer 2 (**Point 51**), we have revised "four typical landscape types" to "four landscapes". Please see lines 11-12 in revision version.

Point 21: L14, ff. change "tomato planting fields" to "tomato fields"

Response: We agree with the comment. We have revised "tomato planting fields" to "tomato fields" in our whole manuscript. Please see lines 14-15, 17-18, 22, 149-150, 184, 220, 224, 235, 257-258, 259-260, 264, 377, 402, 412, 418 and 423 in revision version.

Point 22: L15-20. There are problems with this long sentence. At L17 the verb is missing. At L19 the flower fields landscape is mentioned but providing two different density values and a "respectively", which suggest that another landscape should have been also mentioned in the sentence. Or those were the densities of nymphs and adults? Please rephrase.

Response: We agree with the comment. We have rephrased those sentences to make it clearer. Please see lines 16-21 in revision version.

Point 23: L22 if you say “were the highest” you can remove “than that in the other landscape types”.

Response: We agree with the comment. We have removed “than that in the other landscape types”. Please see lines 24 in revision version.

Point 24: L23 again there are two density values and a respectively but when referring only to a single habitat type (flower landscapes).

Response: We agree with the comment. We have rephrased this sentence to make it clearer. Please see lines 25 in revision version.

Point 25: L24 the middle position of the plant? or what? this is not specified.

Response: The middle position is the middle position of the tomato plant. We have added relevant mention to make this sentence clearer. Please see lines 26 and 28 in revision version.

Point 26: L39-40 I would remove this sentence because the agricultural landscapes are variable and much depends on the scale you consider. I think that it is reasonable to start the paper with the problem of agricultural intensification that you have presented well at L40-43. You could also consider talking more generally, because what it’s true for China holds, unfortunately, also for many other countries. Perhaps later you can get more into the specific situations of China.

Response: We agree with the comment. We have improved the introduction both the structure and the rationale and start the paper with the problem of agricultural intensification. Please see lines 59-73 in revision version.

Point 27: L43 in agroecosystems

Response: We agree with the comment. We have revised “in the agroecosystem” to “in agroecosystems”. Please see lines 62 in revision version.

Point 28: L44 change “will” to “can”

Response: We agree with the comment. We have revised “will” to “can”. Please see lines 63 in revision version.

Point 29: L45 is “population stability” necessary? I think this is redundant with the concept of population dynamics.

Response: We agree with the comment. We have removed “population stability”. Please see lines 63 in revision version.

Point 30: L50 which supports pest control.

Response: We agree with the comment. We have revised “which is beneficial for the reproduction of natural enemies in pest control” to “which supports pest control”. Please see lines 69 in revision version.

Point 31: L54-56 The justification of the rationale is weak. Under which scenarios pests and more favoured than natural enemies? Which environmental factors did you refer to? Why was the study done in tomato plantations and not in another crop?

Response: This study focused on the effects of different landscape types on the population dynamics and density of *B. tabaci*. Landscape types can directly affect *B. tabaci*, or indirectly affect *B. tabaci* by affecting natural enemies. Natural enemies are one of the many factors that affect the population of *B. tabaci* in different landscape types. Therefore, natural enemies are not taken as the research object in our study. We have rephrased relevant description in our manuscript, please see lines 63-73 in revision version.

In our previous study, we found that the damage degrees of *B. tabaci* in tomato planting fields were different in different landscape types around Kunming, Yunnan Province. Therefore, we hypothesized that the different agricultural landscape types may affect the dynamics and densities of whitefly in tomato planting fields. Please see lines 82-85 in revision version.

Point 32: L74-75 unclear, please rephrase. Which landscape variables were considered?

Response: We agree with the comment. We have rephrased this sentence to make it clearer. Please see lines 106-136 in revision version.

Point 33: L89 both years? which years?

Response: We have rephrased this sentence. The study was carried out in tomato planting fields in 2018 and 2019 under different landscape types. Please see lines 143 in revision version.

Point 34: L94 this sentence does not make much sense unless you specify how was the sampling done instead of saying “by sampling”.

Response: We agree with the comment. We have rephrased this sentence to make it clearer. Please see lines 148 in revision version.

Point 35: L123 “Four spikes of tomato were reserved in the tomato plants in the fields.” What was the purpose of this? Perhaps there is nothing wrong with it, but could you check the expression “spike of tomato”?

Response: Peasant household around Yunnan province usually reserve three or four spikes when planting tomatoes. To make the cultivation and management of tomato fields consistent in different landscapes, we reserved four spikes of tomatoes in our study. We confirm that “spike of

tomato” is correct.

Point 36: L126 to be weighed

Response: We agree with the comment. We have revised “to weigh” to “to be weighed”. Please see lines 182 in revision version.

Point 37: L131 were compared

Response: We agree with the comment. We have revised “was compared” to “were compared”. Please see lines 191 in revision version.

Point 38: L132 densities of different positions? On the plant you mean? Unclear.

Response: The different positions are the position of the tomato plant. We have added relevant mention to make this sentence clearer. Please see lines 192 in revision version.

Point 39: L135 change “made by” to “create using”

Response: We agree with the comment. We have revised “made by” to “create using”. Please see lines 197 in revision version.

Point 40: L140 accounted for

Response: We agree with the comment. Combined with the comment (**Point 12**), we have removed this sentence and supplemented interpretation of the PCA in the methods. Please see lines 113-136 in revision version.

Point 41: L146 rivers

Response: We agree with the comment. We have revised “river” to “rivers”. Please see lines 128 in revision version.

Point 42: L149 mountains

Response: We agree with the comment. Combined with the comment (**Point 12**), we have removed this sentence and supplemented interpretation of the PCA in the methods. Please see lines 131-133 in revision version.

Point 43: L151-152 Three of them are divided into urban landscape type, their main landscape elements are urban. What does it mean that they are “urban”?

Response: Urban landscape types were close to the town and their main landscape cover types were buildings. Please see lines 134-136 in revision version.

Point 44: L158 was longer than

Response: We agree with the comment. Combined with the comment from Referee 2 (**Point 72:** L177 it is better to use “as compared to” instead of “than that in the” in whole results sections), We have revised “was the longest than that in the” to “was longer as compared to”. Please see lines 219 in revision version.

Point 45: L159 Aug. if you want to be consistent, but I think later on it should be Sept.

Response: We agree with the comment. We have revised “Aug” to “Aug.”. Please see lines 221 in revision version. And we have revised “Sep.” to “Sept.” in our whole manuscript. Please see lines 222, 225, 228, 237, 240, 242-243, 250, 252, 263, 269, 272, 276, 278, 284-285 and 751-752 in revision version.

Point 46: L164-165 “The main activity period of *B. tabaci* nymphs in the same period” is confusing. Could you please rephrase?

Response: We agree with the comment. We have rephrased this sentence to make it clearer. Please see lines 226-227 and 241 in revision version.

Point 47: L203-204, 227-228 “regardless of the planting years” means “in both years”?

Response: The study was carried out in tomato planting fields in 2018 and 2019. We have rephrased this sentence to make it clearer. Please see lines 261-262 and 280-281 in revision version.

Point 48: L290-292 Please revise the following sentence: “The yield of tomato in the tomato planting fields in the flower landscape types, more than 110 000 kg / ha”. Avoid unnecessary repetition, if you say ‘tomato yield’, clearly, it is from the tomato fields. Do not forget the verb.

Response: We agree with the comment. We have revised this sentence to make it clearer. Please see lines 356-358 in revision version.

Point 49: L327 remove “kind of”

Response: We agree with the comment. We have removed “kind of”. Please see lines 402 in revision version.

Reviewer: 2

Point 50: L1-3 title may be revised as “Regulation of dynamics and densities of Whitefly *Bemisia tabaci* by agricultural landscapes in south China”

Response: We agree with the comment. We have revised the title to “Regulation of dynamics and densities of Whitefly *Bemisia tabaci* by agricultural landscapes in south China”. Please see lines 1-3 in revision version.

Point 51: L12 We selected four typical landscape types (flower field, mountain, river, and urban) as the main elements in Yunnan province, South China.

Response: We agree with the comment. Combined with the comment from Referee 1 (**Point 20**), We have revised this sentence to “We selected four landscapes (flower field, mountain, river, and urban) based on Principal Components Analysis in Yunnan province, South China.”. Please see lines 12-13 in revision version.

Point 52: L35 polyphagous in nature, remove species, remove comma after hosts

Response: We agree with the comments. We have revised “polyphagous” to “polyphagous in nature”, and we have removed species and the comma (,) after hosts. Please see lines 77 in revision version.

Point 53: L36 remove comma after cotton

Response: We agree with the comment. We have removed the comma after cotton. Please see lines 78 in revision version.

Point 54: L36 The nymphs and adults of *B. tabaci* can cause

Response: We agree with the comments. We have revised “Its adults and nymphs can cause economic damage” to “The nymphs and adults of *B. tabaci* can cause”. Please see lines 78-79 in revision version.

Point 55: L37 economic damage directly by sucking plant fluids and indirectly by transmitting plant viruses including

Response: We agree with the comments. We have revised “economic damage by sucking plant fluids and transmitting plant viruses including” to “economic damage directly by sucking plant fluids and indirectly by transmitting plant viruses including”. Please see lines 79-81 in revision version.

Point 56: L39-40 urban area,

Response: Combined with the comment from Referee 1 (**Point 26**), We have removed this sentence.

Point 57: L44 The change in agricultural landscape pattern

Response: We agree with the comments. We have revised “The change of agricultural landscape pattern” to “The change in agricultural landscape pattern”. Please see lines 63 in revision version.

Point 58: L67 conducive in decreasing

Response: We agree with the comments. We have revised “conducive to decreasing” to “conducive in decreasing”. Please see lines 96 in revision version.

Point 59: L89-90 please mentioned the years of study

Response: We agree with the comments. We have mentioned the years of study (2018 and 2019). Please see lines 143 in revision version.

Point 60: L91 against insect or disease pests in our experiments

Response: We agree with the comments. We have revised “against pests or diseases in our experiments” to “against insect or disease pests in our experiments”. Please see lines 145 in revision version.

Point 61: L97 Leaves with *B. tabaci* nymphs and adults

Response: We agree with the comments. We have revised “Leaves with *B. tabaci* adults and nymphs” to “Leaves with *B. tabaci* nymphs and adults”. Please see lines 151-152 in revision version.

Point 62: L98 using the fixed five points sampling method

Response: We agree with the comments. We have revised “using the same five points sampling method” to “using the fixed five points sampling method”. Please see lines 152-153 in revision version.

Point 63: 101 remove coma (,) before and

Response: We agree with the comment. We have removed the comma before and. Please see lines 156 in revision version.

Point 64: L102 observed per plot

Response: We agree with the comments. We have revised “monitored per plots” to “observed per plot”. Please see lines 157 in revision version.

Point 65: L103 remove coma (,) before and

Response: We agree with the comment. We have removed the comma (,) before and. Please see lines 158 in revision version.

Point 66: L112 remove coma (,) after main

Response: We agree with the comment. We have removed the comma after main. Please see lines 167 in revision version.

Point 67: L112 and end for each of these,

Response: We agree with the comments. We have revised “and end of each of these” to “and end for each of these”. Please see lines 167-168 in revision version.

Point 68: L113-118 must be in past sentences

Response: We agree with the comments. We have revised those sentences into the past tense. Please see lines 168-174 in revision version.

Point 69: L134 $P=0.05$

Response: We agree with the comments. We have revised “ $P=0.05$ ” to “ $P=0.05$ ”. and we have revised this sentence to make it clearer. Please see lines 193-195 in revision version.

Point 70: L135 figures of population dynamics and cumulative seasonal activity curves

Response: We agree with the comments. We have revised “figures of cumulative seasonal activity curves and population dynamics” to “figures of population dynamics and cumulative seasonal activity curves”. Please see lines 195-197 in revision version.

Point 71: L167 write full September

Response: We agree with the comment. We have written full September. Please see lines 228 in revision version.

Point 72: L177 it is better to use “as compared to” instead of “than that in the” in whole results sections

Response: We agree with the comment. We have revised “than that in the” to “as compared to” in our whole manuscript. Please see lines 219, 225, 234, 239, 293, 296, 308, 314, 327, 332, 338, 346 and 380 in revision version.

Point 73: L266 better use 146.88 for F and 0.001 for P

Response: We agree with the comment. The value of F was corrected to two decimal places and the value of P was corrected to three decimal places in our whole manuscript. Please see lines 297-298, 301, 309, 316-317, 321, 325, 328, 333-334, 339-345, 348-354 and 360-361 in revision version.

Appendix B

Dr Polly Campbell (Associate Editor) and Pete Smith (Subject Editor)
Editor, Royal Society Open Science

25 Feb 2022

Dear Dr Polly Campbell (Associate Editor) and Pete Smith (Subject Editor),

We are receipt of the set of comments by reviewers, and we thank you for the fast and efficient handling of our manuscript. We are happy to read that you found our MS is greatly improved. We also thank the reviewers for provided valuable suggestions, each of which we have carefully considered. We added our replies after each comment separately – please find these below.

As a result, we revised our manuscript that we hope it is now easier to be read and comprehended. We hope you will find our revision satisfactory, and our MS can be accepted for publication in Royal Society Open Science.

Again, thank you for your editorial assistance, and we await further from you concerning our manuscript.

Sincerely yours

Dr. Xiaoming Zhang

Corresponding author

Associate Editor Comments to Author (Dr Polly Campbell):

The revised version of this manuscript is greatly improved and the authors' careful attention to detail is appreciated. Please address all the reviewer's minor comments before final submission.

Response: Thanks very much for your affirmation to our manuscript. We also thank the reviewer for provided valuable suggestions, each of which we have carefully considered. We added our replies after each comment separately. And we revised the abstract to make it within the word count limit (200 words), Please see lines 10-30 in revision version.

Reviewer comments to Author:

Reviewer: 3

Point 1: Line 1 is Whitefly, but the correct one is whitefly

Response: We agree with the comment. We have revised "Whitefly" to "whitefly". Please see lines 1-2 in revision version.

Point 2: Line 13 the correct one is Yunnan Province, south China

Response: We agree with the comment. We have revised "Yunnan province, South China" to "Yunnan Province, south China". Please see lines 13 in revision version.

Point 3: Line 16 is density, but the correct one is densities

Response: We agree with the comment. We have revised "density" to "densities". Please see lines 16 in revision version.

Point 4: Line 41-43 These diverse noncrop habitats can provide foods, shelters, alternative hosts, and favorable microclimates for natural enemies in the field, which supports pest control.

Response: We agree with the comment. We have revised this sentence to "These diverse noncrop habitats can provide foods, shelters, alternative hosts, and favorable microclimates for natural enemies in the field, which supports pest control.". Please see lines 43-47 in revision version.

Point 5: Line 51-52 add Abd-Rabou S and Simmons AM. Survey of reproductive host plants of *Bemisia tabaci* (Hemiptera: Aleyrodidae) in Egypt, including new host records [J]. *Entomological News*, 2010, 121(5): 456-465.

Response: We agree with the comment. Through intensive reading this paper, we believe that this paper can support our point of view more than the former cited literature, so we have revised this sentence and replace the former cited literature [18] with this literature. Please see line 55 and 428-432 in revision version.

Point 6: Line 60, 62, 67, 314 Province

Response: We agree with the comment. We have revised “province” to “Province”. Please see lines 64, 66-67, 71 and 318 in revision version.

Point 7: Line 65 delete comma (,) after river

Response: We agree with the comment. We have deleted comma (,) after river. We also revised the same errors in the full manuscript. Please see line 12 and 69 in revision version.

Point 8: Line 78 south China

Response: We agree with the comment. We have revised “South China” to “south China”. Please see lines 82 in revision version.

Point 9: Line 96 is main types, but the correct one is main species

Response: We agree with the comment. We have revised “main types” to “main species”. Please see lines 100 in revision version.

Point 10: Line 99 is rivers, but the correct one is river

Response: We agree with the comment. We have revised “rivers” to “river”. Please see lines 103 in revision version.

Point 11: Line 135 delete comma (,) after 50

Response: We agree with the comment. We have deleted comma (,) after 50. Please see line 139 in revision version.

Point 12: Line 156 delete comma (,) after main

Response: We agree with the comment. We have deleted comma (,) after main. Please see line 160 in revision version.

Point 13: Line 166-168 In 2018, the main activity period of *B. tabaci* nymphs in the flower landscape type, as long as 52 days, was longer as compared to other landscape types. The peak activity date in the tomato fields in the river landscape type, 28 Aug.,.....

Response: We agree with the comment. We have revised this sentence to “In 2018, the main activity period of *B. tabaci* nymphs in the flower landscape type, as long as 52 days, was longer as compared to other landscape types. The peak activity date in the tomato fields in the river landscape type, 28 Aug.,.....”. Please see lines 170-172 in revision version.

Point 14: Line 170 were later relative to.....

Response: We agree with the comment. We have revised “was later relative to” to “were later relative to”. Please see lines 174 in revision version.

Point 15: Line 173 main activity periods

Response: We agree with the comment. We have revised “main activity period” to “main activity periods”. Please see lines 177 in revision version.

Point 16: Line 175 were in early September.

Response: We agree with the comment. We have revised “was in early September” to “were in early September”. Please see lines 179 in revision version.

Point 17: Line 181-182, 192, 196, 202 in the flower landscape type

Response: We agree with the comment. We have revised “in the flower landscape types” to “in the flower landscape type”. We also revised the same errors in the full manuscript. Please see lines 29-30, 185-186, 195-196, 200 and 206 in revision version.

Point 18: Line 183, 226, 261 in the urban landscape type

Response: We agree with the comment. We have revised “in the urban landscape types” to “in the urban landscape type”. Please see lines 187, 230 and 264-265 in revision version.

Point 19: Line 189 and their peak activity date were in mid-Sept..

Response: We agree with the comment. We have revised “and its peak activity date was in mid-Sept.” to “and their peak activity date were in mid-Sept.”. Please see lines 193 in revision version.

Point 20: Line 198, 253, 264, 304, 321 in the river landscape type

Response: We agree with the comment. We have revised “in the river landscape types” to “in the river landscape type”. We also revised the same errors in the full manuscript. Please see lines 21, 202, 257, 268, 308 and 325 in revision version.

Point 21: Line 219 In the mountain landscape type,

Response: We agree with the comment. We have revised “In the mountain landscape,” to “In the mountain landscape type,”. Please see lines 223 in revision version.

Point 22: Line 255 delete (, were the highest relative to the other landscape types)

Response: We agree with the comment. We have deleted (, were the highest relative to the other landscape types). Please see line 259 in revision version.

Point 23: Line 255-256 At each activity period

Response: We agree with the comment. We have revised “At each activity periods” to “At each activity period”. Please see lines 259-260 in revision version.

Point 24: Line 271 At the early and late activity period, the densities of male adult *B. tabaci* were.....

Response: We agree with the comment. We have revised this sentence to “At the early and late activity period, the densities of male adult *B. tabaci* were.....”. Please see lines 275-276 in revision version.

Point 25: Line 273 At the main activity period

Response: We agree with the comment. We have revised “At the main activity periods” to “At the main activity period”. Please see lines 276-277 in revision version.

Point 26: Line 279-281 The density of *B. tabaci* nymphs in the middle position of the tomato plants was significantly higher as compared to other positions, regardless of landscape types

Response: We agree with the comment. We have revised this sentence to “The density of *B. tabaci* nymphs in the middle position of the tomato plants was significantly higher as compared to other positions, regardless of landscape types”. Please see lines 283-285 in revision version.

Point 27: Line 286-287 Additionally, the density of adult *B. tabaci* in the upper position of the tomato plants was higher as.....

Response: We agree with the comment. We have revised this sentence to “Additionally, the density of adult *B. tabaci* in the upper position of the tomato plants was higher as.....”. Please see lines 290-291 in revision version.

Point 28: Line 296-298 The yield of tomato were different under different landscape types. The yield of tomato in the flower landscape type were more than 110 000 kg / ha in both years, which were the highest.....

Response: We agree with the comment. We have revised this sentence to “The yields of tomato were different under different landscape types. The yield of tomato in the flower landscape type were more than 110 000 kg / ha in both years, which were the highest.....”. Please see lines 300-302 in revision version.

Point 29: Line 331 After the scientific names mentioned for the first time, the names of author, order and family must be placed.

Response: We agree with the comment. We have placed the author, order and family after the scientific names mentioned for the first time. Please see lines 335-336 in revision version.

Point 30: Line 336 in the middle position

Response: We agree with the comment. We have revised “in the middle positions” to “in the middle position”. Please see lines 341 in revision version.

Point 31: Line 463 Lövei GL

Response: We agree with the comment. We have revised the author`s name to “Lövei GL”. Please see lines 470 in revision version.